# A machine learning approach to address air quality changes during the COVID-19 lockdown in Buenos Aires, Argentina

Melisa Diaz Resquin[1,2,3,*], Pablo Lichtig[1,4], Diego Alessandrello[1], Marcelo De Oto[1], Darío Gómez[1,2], Cristina Rössler[1,5], Paula Castesana[1,4,6], and Laura Dawidowski[1,5,*]

[1]Comisión Nacional de Energía Atómica, Gerencia Química, Buenos Aires, Argentina.
[2]Facultad de Ingeniería, Universidad de Buenos Aires, Buenos Aires, Argentina.
[3]Center for Climate and Resilience Research (CR)[2], Santiago, Chile.
[4]Consejo Nacional de Investigaciones Científicas y Técnicas, Buenos Aires, Argentina.
[5]Instituto de Investigación e Ingeniería Ambiental, Universidad Nacional de San Martín, Buenos Aires, Argentina.
[6]YPF Tecnología S.A. (Y-TEC), Buenos Aires, Argentina.
[*]These authors contributed equally to this work.

**Correspondence:** Melisa Diaz Resquin (mdiazresquin@fi.uba.ar), Laura Dawidowski (dawidows@cnea.gov.ar)

**Abstract.**

Having a prediction model for air quality at a low computational cost can be useful for research, forecasting, regulatory, and monitoring applications. This is of particular importance for Latin America, where rapid urbanization has imposed an increasing stress on the air quality of almost all cities. In recent years, machine learning techniques are being increasingly accepted as a useful tool for air quality forecasting. Out of these, Random Forest has proven to be an approach that is both well-performing and computationally efficient while still providing key components reflecting the non-linear relationship among emissions, chemical reactions, and meteorological effects. In this work, we employed the Random Forest methodology to build and test a forecasting model for the city of Buenos Aires. We used this model to study the deep decline in most pollutants during the lockdown imposed by the COVID-19 (COronaVIrus Disease 2019) pandemic, by analyzing the effects of the change in emissions while taking into account the changes in the meteorology, using two different approaches. First, built Random Forest models trained with the data from before the beginning of the lockdowns. We used it to make predictions of the business-as-usual scenario during the lockdowns, and estimated the changes in concentrations by comparing the model results with the observations. This allowed us to assess the combined effects of the particular weather conditions and the reduction in emissions during the period when restrictions were in place. Second, we used Random Forest with meteorological normalization to compare the observational data from the lockdown periods with the data from the same dates of 2019, decoupling the effects of the meteorology from short-term emission changes. This allowed us to analyze the general effect that restrictions similar to those imposed during the pandemic could have on pollutant concentrations, and that information could be useful to design mitigation strategies.

The results during testing showed that the model captured the observed hourly variations and the diurnal cycles of these pollutants with a normalized mean bias of less than 6% and Pearson correlation coefficients of the diurnal variations of between 0.64 and 0.91 for all the pollutants considered. Based on the Random Forest results, we estimated that the lockdown implied relative changes in concentration of up to -45% for CO, -75% for NO, -46% for $NO_2$, -12% for $SO_2$ and -33% for $PM_{10}$

during the strictest mobility restrictions. $O_3$ had a positive relative change in concentration (up to an 80%), that is consistent with the response in a VOC-limited chemical regime to the decline in $NO_x$ emissions. The relative changes estimated using the meteorological normalization technique show mostly smaller changes than those obtained by the Random Forest predictive model. The relative changes were up to -26% for CO, up to -47% for NO, -36% for $NO_2$, -20% for $PM_{10}$ and up to 27% for $O_3$. $SO_2$ is the only species that had a larger relative change when the meteorology is normalized, up to 20%. This points out the need for accounting not only for differences in emissions, but also in meteorological variables in order to evaluate the lockdown effects on air quality. The findings of this study may be valuable for formulating emission control strategies that do not disregard their implication on secondary pollutants. We believe that the model itself can also be a valuable contribution to a forecasting system in the city, and that the general methodology could also be easily applied in other Latin American cities as well. We also provide the first $O_3$ and $SO_2$ observational dataset in more that a decade for a residential area in Buenos Aires, openly available at https://data.mendeley.com/datasets/h9y4hb8sf8/1 (Diaz Resquin et al., 2021).

## 1  Introduction

In recent times, machine learning has been proven to be an efficient approach for air quality prediction by relying on historical data to estimate the temporal variability of different pollutants for a specific site at a low computational cost. Also, this kind of model has the ability to unravel underlying patterns in data and deal with complex interactions among predictive variables (Stafoggia et al., 2020).

During the last decade, random forest rose as a new method for the prediction of mean values of atmospheric pollutants (Yu et al., 2016; Feng et al., 2019; Jiang and Riley, 2015). This is a supervised machine learning method, consisting of applying multiple tree classifiers created at random using bagging (i. e., selecting samples stochastically to create new datasets, of which every classification tree is created). RF requires a short training time and can provide reliable information on air quality, with a strong anti-overfitting ability (Liu et al., 2021). Many data science programming languages have libraries where random forest is already efficiently implemented (e. g., `scikit-learn` in Python or `randomForest` in R). Random forest is faster and cheaper than other available models, such as regional chemical transport models (CTMs), in terms of computation costs, needs less input variables and is a useful method when information on air pollutant concentrations at a particular site is needed. According to Masih (2019), machine learning techniques may even provide better forecasting than CTMs and, out of the different existing algorithms, random forest seems to stand out due to its simplicity and the quality of its results, which can account for non-linear relationships between emissions, chemical reactions, and meteorological effects. With respect to complex reactive species, the random forest method has also been successfully used to assess $O_3$ levels. For example Zhan et al. (2018) satisfactorily applied the random forest method to predict spatio-temporal variability of daily $O_3$ concentrations across China using information on meteorology, elevation and emission inventories. One of the most recent applications of machine learning methods has been aimed at elucidating the interlinkage among the COVID-19 pandemic lockdown measures, human mobility and air quality (Rahman et al., 2021; Velders et al., 2021; Yang et al., 2021).

The outbreak of the COVID-19 pandemic at the end of 2019, with its devastating consequences in terms of loss of life and economic impact, has caused many governments around the world to impose different degrees of lockdown. For atmospheric scientists, it has also provided a unique opportunity to examine changes in air pollution under decreased emission levels, in what Gaubert et al. (2021) called an unintentional worldwide experiment. Many studies have, in general, identified significant decreases of most pollutants, except for $O_3$, under the stay-at-home orders imposed against COVID-19 (Muhammad et al., 2020; Faridi et al., 2021; Srivastava, 2021; Grange et al., 2021; Yang et al., 2021). These drastic changes in anthropogenic emissions are of major interest to enhance our understanding of the chemistry related to air quality, particularly when the behavior of secondary pollutants, like ozone ($O_3$) or components of particulate matter (PM), is explored (Gaubert et al., 2021). $O_3$, in particular, has a complex behavior depending on multiple factors. Nitrogen monoxide (NO) and nitrogen dioxide ($NO_2$) conform $NO_x$, that together with volatile organic compounds (VOCs) play vital roles in the $O_3$ formation process, and its production can be either VOC-limited or $NO_x$-limited (Shi and Brasseur, 2020; Liu et al., 2021; Li et al., 2019).

An early approach to analyze the changes in air quality due to the implementation of specific control measures was to comparatively assess concentrations during the lockdown with concentrations of the same period of the previous year or the mean value of a period of five years, using exclusively ground-based or satellite observations. However, the degree to which the COVID-19 lockdown influenced air quality is not only a function of emissions, but also of both meteorology and physical and chemical atmospheric transformations (Kroll et al., 2020; Le et al., 2020). In consequence, pure statistical tests or observational comparisons might be inadequate to have a complete understanding of what influences pollutant concentrations, since weather conditions, particle persistence, transport, radiation and seasonality affect concentrations by linear and non-linear processes (Šimić et al., 2020). In this work, this challenge has been addressed using two different, but complementary approaches. The first one consists in using a model to simulate a hypothetical scenario in which the restrictions were not implemented, which we did using the random forest algorithm (RF), as previously done by Velders et al. (2021). The second one consists in a random forest based normalization of the meteorological variables, which makes it possible to decouple the emission changes (Shi et al., 2021; Grange and Carslaw, 2019; Vu et al., 2019).

The goals of this study were: (i) to provide novel air quality data for the Metropolitan Area of Buenos Aires (MABA), Argentina, including the first $O_3$ and $SO_2$ observational datasets in a residential area in more than a decade; (ii) to explore the performance of the random forest method in predicting the air quality situation at two monitoring sites of the MABA; (iii) to apply this methodology to estimate the changes in air pollutant concentrations under the COVID-19 control measures; and (iv) to assess the effect of the reduction on emissions by normalizing the meteorological variables. We implemented the RF algorithm to estimate the concentrations of CO, $NO_2$, NO, sulfur dioxide ($SO_2$), $O_3$, and particles with aerodynamic diameter less or equal than 10 $\mu m^3$ ($PM_{10}$) using meteorological and air quality observations, as well as the local diurnal variation of emissions as explanatory variables. Trained with data acquired in 2019 and 2020 before the start of the pandemic with the variables available for this city, the RF method can only predict concentrations under a business-as-usual (BAU) scenario. We then compared this BAU estimations with the observations during two distinct lockdown phases. We also used a random forest normalization technique (RFN) to decouple the effects of the meteorology over the concentration of the pollutants by normalizing the meteorological variables based on Shi et al. (2021). We compared them with the normalized observations

for the same period of the the previous year, allowing us to assess the effect of reducing the emissions, independently of the particular meteorological situation that occurred during the specific periods analyzed. In addition, we studied the responses of $O_3$ to the reduction in emissions of its precursors ($NO_x$ and VOCs) because of its relevance regarding emission control and health effects.

The remainder of this paper is structured as follows. Section 2 provides a description of the studied area, the different lockdown phases, the air quality and meteorological data and the structure of the random forest models used to estimate the relative changes (RC) during the lockdown. The analysis of the models performance and the analysis of the impact due to the emission reductions are in Section 3. Section 4 provides a description of the data and code availability. Finally, Section 5 presents a summary and the main conclusions of this work.

## 2 Material and methods

### 2.1 Description of the studied area

The MABA comprises the Autonomous City of Buenos Aires (ACBA) and 40 surrounding Districts of the Greater Buenos Aires (GBA). Located along the western coast of the Río de la Plata estuary, on a flat plain, the MABA is the third biggest Megalopolis of Latin America and the Caribbean. It has a population of approximately 13 millions, with a heterogeneous population density in the 14–20 thousands $\mathrm{inhab\,km^{-2}}$ range. Its active fleet reached 5.4 million vehicles by 2019 (Anapolsky, 2020).

In terms of anthropogenic air pollutant emissions road transportation is clearly the largest contributor of CO, VOCs and PM in the area. The MABA is also affected by the emissions from residential, commercial and institutional buildings, mainly based on natural gas consumption, and from three power plants, located near the shoreline of the La Plata River, which mainly burn natural gas and, to a lesser extent, gas oil and fuel oil. Under these circumstances, $NO_x$ is emitted by stationary and mobile sources in a similar amount (Castesana et al., 2021). Since most of Buenos Aires' vehicle fleet uses low-sulfur fuel, the majority of $SO_2$ emissions are due to heavy duty diesel engines, used by ships, trucks and, occasionally, small electricity generators.

### 2.2 Description of the lockdown for the MABA

Argentina's national government established different lockdown phases for the duration of the pandemic (Decree 297/2020, 2020). Since 80% of Argentina's COVID-19 cases were concentrated in the MABA, some policies applied to the MABA region differed from those applied to the rest of the country. Starting on 20 March 2020, strict measures were imposed to avoid a sharp increase in COVID-19 cases, emphasizing that the population should stay at home and avoid any social contact. All non-essential stores, including toys, furniture and clothing stores, were closed until 11 May. Table 1 provides a summary of the restrictions set for the MABA during each phase. Under severely restricted mobility, public transport and passenger car circulation decreased drastically. Local mobility dropped down 80% during the Intense lockdown phase and 65% for the

120 Flexible lockdown phase until the end of May (Aktay et al., 2020). It is worth noting that, before the COVID-19 pandemic, 1 million vehicles entered the city of Buenos Aires from the suburbs per day.

Considering the different degrees of the restrictions imposed, we evaluated the impact of the lockdown on air quality according to two distinct periods. The first period, from 20 March to 12 April 2020, corresponded to the most restrictive lockdown (LD). The second period, from 13 April to 25 May, was denominated partial lockdown (PLD) because some restrictions were 125 lifted. The period 1–15 March 2020, before the start of the first lockdown, was defined as BLD and was used to evaluate the model. As from 16 March, flexible restrictions started, but were optional, therefore the period 16–19 March was not considered in our research.

Being combustion the main air pollution source in the area, the significant decrease in traffic flow imposed by the lockdown led necessarily to a decrease in the emissions of traffic-related pollutants (D'Angiola et al., 2010; Puliafito et al., 2017; Diaz 130 Resquin et al., 2018; Castesana et al., 2021).

### 2.3 Meteorological description

The atmospheric general circulation in the MABA is controlled by the influence of the semi-permanent South Atlantic High pressure system. This system influences the climate of the MABA throughout the year by bringing in moist winds from the northeast, which produce most of the precipitation in the area in the form of frontal systems, or storms produced by 135 cyclogenesis, in autumn and winter (Barros et al., 2006). In terms of the climate conditions of the MABA, temperatures at the beginning of autumn range from warm to hot in the afternoon, but they are mild in the nights and the mornings. Later on in the season, conditions are cooler, featuring mild afternoons, and cold nights and mornings.

To identify similarities and differences between the meteorological conditions during the lockdown phases and the testing period (BLD, LD and PLD) with those of the autumn of 2019 (March, April and May, MAM2019) we carried out a meteo-140 rological analysis for all the periods. We used hourly and daily data from the Buenos Aires Central Observatory (OBS: Lat: 34º 35' S Lon: 58º 29' W). The site of the Meteorological Weather Service of Argentina is located in a residential area. It is representative of the meteorology of the air quality conditions under study.

Average temperatures in the BLD (24.4 °C) and in the LD (21.1 °C) were higher than that in MAM2019 (18 °C) while the average temperature in the PLD (16.8 °C) was lower than that in MAM2019, but close to the corresponding value in May 2019 145 (16 °C). Precipitation in March and April 2020 exceeded the accumulated values of the same months of 2019 (+60% and + 90% respectively). On the contrary, precipitation in May 2020 exhibited significantly lower values than those of 2019 (-75%).

During MAM2019, the average calm value was 6.7%, while during the BLD, the LD and the PLD, the corresponding calm values were 3.6%, 4.7% and 8.6%. Average wind velocity, within the 7.5–8.6 $km\,h^{-1}$ range, was similar in all periods. In autumn 2020, the prevailing wind was from the NW-N sector with an average contribution of 34% against 26.5% in 2019. The 150 LD and the PLD periods had a similar direction of prevailing winds as autumn 2019, contrarily 45% of winds during the BLD were from the NE–E sector.

**Table 1.** Description of the lockdown phases on the MABA. NU Not used (Not included in the model).

| Period (2020) | Phase | Denomination | Description | Mobility |
|---|---|---|---|---|
| 1 – 15 Mar. | Before the lockdown | BLD | Pandemic had started in South America, but no restrictions were thus far implemented in Argentina. | 100% |
| 16 – 19 Mar. | School Closedown and Optional Lockdown | NU | Countrywide, all schools and universities were closed. People were advised to stay at home. Theaters and cinemas were shut. Public events over 200 people were cancelled. Implementing home-office was recommended. Gatherings were to be avoided. | 90% |
| 20 – 30 Mar. | Strict Lockdown | LD | Bars, restaurants, shopping centers and stores in general were closed, with the exception of food stores and pharmacies. Only essential economic activities were permitted. Circulation of passenger vehicles was only allowed with a special permit. Public transport was limited within the region. Most industrial activities were suspended. Only groceries were allowed to be delivered. These restrictions apply countrywide, regardless of the amount of cases informed. The country and the district borders were closed. | 20% |
| 31 Mar. – 12 Apr. | Flexible Lockdown I | LD | Food delivery was permitted. | 20% |
| 13 Apr. – 5 May | District Differentiated Lockdown I | PLD | More economic activities were permitted. More stores were permitted to open in several districts. The lockdown in the MABA continued, but people started to be less careful about the social distancing measures. | 35% |
| 6 – 25 May | District Differentiated Lockdown II | PLD | All non-essential stores, including toys, furniture and clothing stores, were permitted to open with specific protocols. Children were allowed to go for a walk with an accompanying adult on the weekends, but no farther than 500m from home. | 35% |

Our analysis showed that there were meteorological differences in terms of temperature and precipitation between autumn 2019 and the periods analyzed in 2020 (BLD, LD, and PLD). This is indicative of the need of taking into account the influence of meteorological conditions for comparative purposes of air quality conditions that occurred in the different periods.

 ## 2.4 Air quality data

We employed air quality data from two monitoring sites: Comisión Nacional de Energía Atómica (CNEA), operated by our research group, and Parque Centenario (PC), managed by the Autonomous City of Buenos Aires (described below). Both sites are mostly influenced by the emissions from mobile and residential sources, and, to a lesser extend, by the thermal power plants, located at least at 6 km from them (Diaz Resquin et al., 2018; Pineda Rojas et al., 2020).

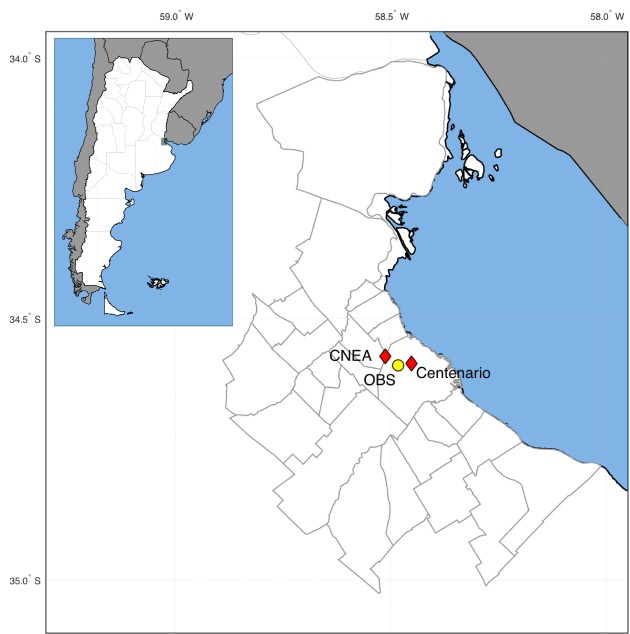

**Figure 1.** Location of MABA in Argentina (top left); zoom of the MABA(right). In yellow, the location of OBS, the site of the National Meteorological Service monitoring site referred in this study, and in red, the air quality monitoring sites (shape files from IGN, 2021).

### 160 2.4.1 Comisión Nacional de Energía Atómica

From 23 February 2019 to 26 May 2020 a monitoring campaign was carried out in an open area (-34.57 ºS, -58.51 ºW) situated 14 km away from the Buenos Aires City center (Figure 1) to assess the levels of different gases (CO, NO, $NO_2$, $SO_2$, and $O_3$) and their temporal variability in a residential area of the MABA.

The main goal of this monitoring campaign was to assess the temporal variability of $SO_2$ and $O_3$ in the area for an entire
165 year. Although it may seem surprising, especially for a megacity like the MABA, there is scarce and fragmentary information on the concentrations of $SO_2$ and $O_3$. Presently, $O_3$ is routinely monitored in one site of the MABA, located in an industrial area. Past data for the region are only available from a few short-time campaigns carried out in the early 2000s (Reich et al., 2006). Similarly, there is a lack of monitored $SO_2$ concentrations because historical measurements carried out in the 1990s reported very low values, and therefore the decision makers decided not to measure this pollutant on a regular basis. However,

**Table 2.** Description of the equipment used in CNEA site

|         | Instrument         | Description                                                                                                          | Calibration*                                    |
|---------|--------------------|---------------------------------------------------------------------------------------------------------------------|-------------------------------------------------|
| CO      | Horiba APMA-370    | Sampler with a non-dispersive infrared absorption photometry sensor with a solenoid valve with cross flow modulation. | 12.44 ppmV                                      |
| $O_3$   | Horiba APOA-370    | Detector that operates with a cross flow modulation, ultra-violet absorption method in conjunction with the comparative calculation method. | 0.1 ppmV. ± 0.5% diluted          |
| $SO_2$  | Horiba APSA-370    | UV Fluorescence detector.                                                                                           | 0.05 ppmV ± 0.6% diluted                        |
| NO, $NO_2$ | Horiba APNA-370 | Cross flow modulation type with reduced chemiluminescence detector.                                                 | 0.099 ppmV ±1.4% diluted (NO)                   |

\* The calibration of the ambient air gases detectors was performed by following U.S. EPA regulations and Horiba standard procedures (see U.S. EPA CFR 40 Part 50, appendixes A1, C, D and F, and the corresponding user manual for the Horiba AP devices). The APMA-370, APSA-370 and APNA-370 were calibrated using EPA certified calibration gases and diluted with an Environics 6103, a NIST traceable mass flow controller dilutor, when needed.

it has now become a pollutant of concern for local authorities, that have recently decided to start monitoring $SO_2$ in two of the four air quality stations of the ACBA in the near future.

Air pollutant concentrations were continuously acquired (Table 2). Monitors were placed at an approximate height of 10 m, and 100 m E from a main traffic artery with a high density of buses, light duty trucks and passenger cars. Another main artery is located 500 m N, having circulation of vehicles including trucks and buses in a low speed stop-and-go pattern. The international airport Jorge Newbery, two thermal power plants, the La Plata river and the port are located within a 19 km radius of the monitoring station.

Data was registered per one minute averages. Unfortunately, from 26 May onwards, restrictions on entering our institute where the monitoring station was located led to the need to suspend the monitoring campaign.

### 2.4.2 Parque Centenario

To include aerosol variations in this analysis and complement the information of CNEA's site, we used $PM_{10}$, CO, NO, and $NO_2$ data from PC station (34.61 ºS, 58.44 ºW), one of the surface air quality sites of the Environmental protection agency of Buenos Aires city (APRA). This site is located in a residential-commercial area with medium vehicular flow and relatively low incidence of stationary sources. A monthly technical report of the hourly-average concentrations registered in PC is available

at APRA website (APRA, 2020). Although the city has three other monitoring stations, at least one of the essential periods
needed for this study was missing in each of them. Therefore, they were not taken into account for this study.

### 2.4.3   Summary of the datasets

Relatively low concentration values for all the analyzed periods, with no exceedances for short term air quality standard for all the pollutants measured (Decree 1074/18, 2018; Act 1356, 2004) were registered in both sites. Air pollutants, except $SO_2$, exhibited well defined diurnal cycles (see Fig. S2 of the Suplemmentary Material).

CO and $NO_x$ patterns were governed by traffic emissions (Figs. S1 and S2 of Supplementary Material), with the maximum values in winter. Annual mean average values of $NO_x$ were $\sim 37$ ppb for both CNEA and PC. Relevant differences in CO were identified, with annual mean levels in PC doubling those measured in CNEA (0.51 ppm versus 0.26 ppm).

$PM_{10}$, which was only measured in PC, had a mean value of $21 \, \mu g \, m^{-3}$, with the maximum values at noon.

With respect to the pollutants that were only measured in CNEA, $SO_2$ maximum concentrations were registered during autumn (April) with monthly averages in the 2–2.9 ppb range. In terms of $O_3$ concentrations, maximum daylight levels were registered during summer. The diurnal cycle presented higher levels during the afternoon and was opposite to those of NO and $NO_2$.

### 2.5   Modeling Approach

We used the machine learning random forest method to: (i) estimate the relative changes during the LD and the PLD phases and (ii) develop a model for air quality forecast for the MABA, at a low computational cost. To this end, two different approaches have been implemented using a random forest algorithm (Figure 2). The first one estimates the hypothetical prospective pollutant concentrations that would have occurred in the MABA under the regular emissions conditions (BAU scenario), with the particular meteorological conditions that occurred during the period analyzed. This model, named as *random forest predictive model* or simply *RF*, has been applied to the LD and the PLD phases to estimate the concentrations if no lockdown measures had been imposed and compare them with the observations during the lockdown phases. This tool could also be used to forecast the air quality situation in the city. The second approach, referred as *RF normalized* or *RFN*, has been designed to decouple the effects of the meteorology by normalizing the meteorological variables, allowing a generalized assessment of the effect of the changes in the emission patterns. This technique has been applied to compare the concentrations of the different lockdown periods to those of the same time frames of 2019 in order to infer the effects of the sudden reduction in emissions during COVID-19 mobile restrictions period. A summarized schematic of the modeling approach can be seen in Fig. 2.

Observations from February 2019 to May 2020 were divided into different groups following the methodology by Grange et al. (2021), using 8710 total data points for CNEA and 9198 for PC. The training of the models was conducted using a random sample of the 80% of the input data from February 2019 to February 2020. The remaining 20% was used as testing ($t$) to choose the model configuration with best statistical metrics. The BLD period (360 data points, see Table 1) was established as a different evaluation period in order to check the adequate performance of the model two weeks before the lockdown

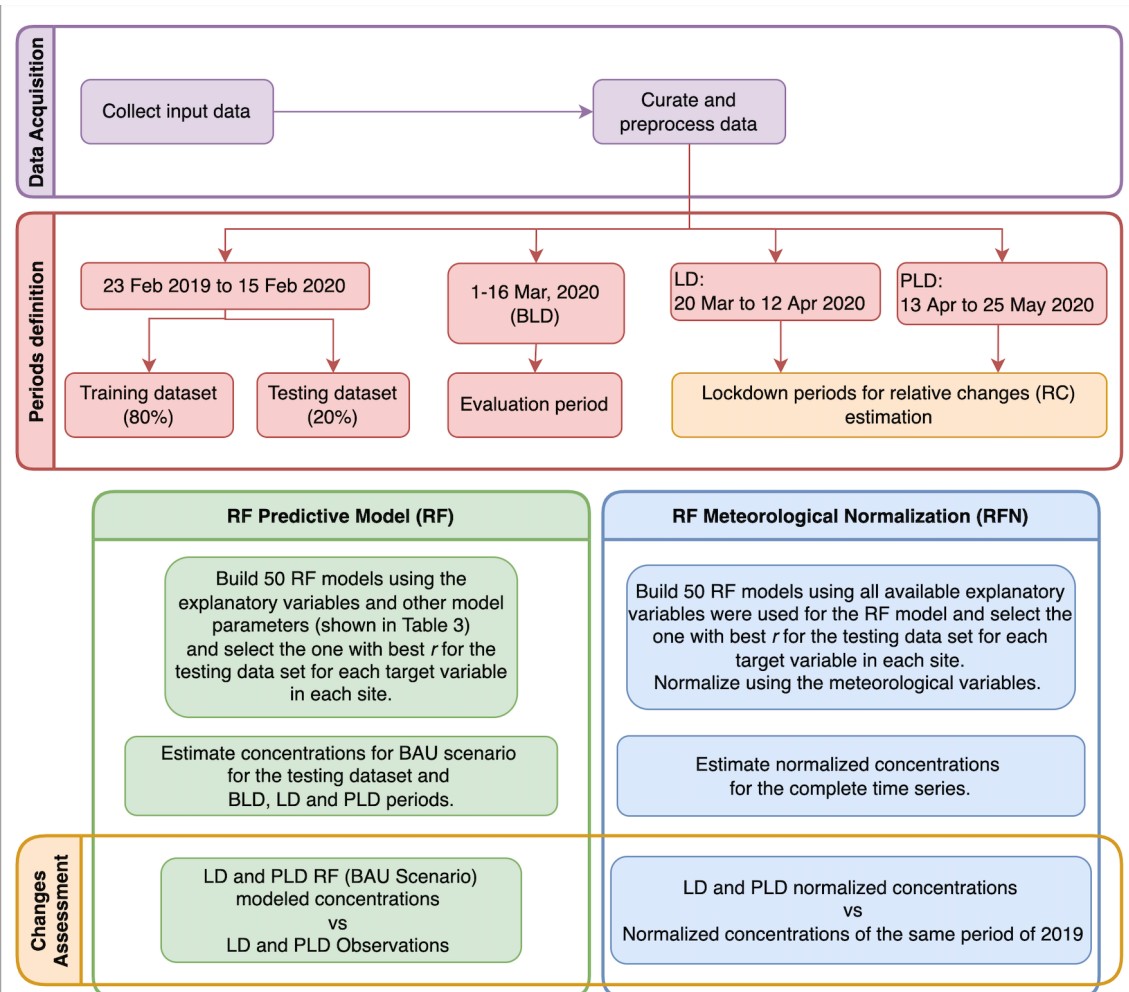

**Figure 2.** Schematic description of model building and evaluation.

periods. Data collected from March 20 to May 25 and the RF estimates were used to quantify and interpret the changes during the LD and the PLD.

The target variables were the measured air pollutant concentrations in each monitoring site, namely CO, NO, $NO_2$, $O_3$, and $SO_2$ (CNEA) and CO, NO, $NO_2$, and $PM_{10}$ (PC).

As predictive variables we considered: (i) data taken from the Meteorological Weather Service, namely wind speed, wind direction, surface temperature, sea level pressure, and relative humidity; (ii) boundary layer height and total cloud cover taken from ERA5 (Hersbach et al., 2020); (iii) the pollutant concentrations measured in each of the sites; (iv) time variables such as month, hour, weekday, and (v) diurnal and weekly emission cycles for pollutants associated with gasoline and diesel emissions (Castesana et al., 2021; Freitas et al., 2011). For the predictive model, all these variables were tested as explanatory variables

**Table 3.** Random Forest model. Target variables, predictors and hyper-parameters for RF.

| Site | Target Variable | Explanatory Variables |
|------|-----------------|------------------------|
| CNEA | CO | t2, ws, wd, blh, gas_emcycle |
| | NO | t2, rh2, slp, ws, wd, blh, tcc, gas_emcycle, aer_emcycle |
| | $NO_2$ | t2 , ws , wd , aer_emcycle,blh, CO, NO |
| | $SO_2$ | t2, rh2, ws, wd, CO, NO, $NO_2$, daynight |
| | $O_3$ | ws, wd, CO, NO, $NO_2$, $SO_2$, daynight |
| PC | CO | t2, ws, wd, blh, gas_emcycle, hour |
| | NO | t2,rh2, slp, ws, wd, blh, gas_emcycle, aer_emcycle, hour |
| | $NO_2$ | t2, U, V, gas_emcycle, aer_emcycle, CO, NO, month, weekday, hour |
| | $PM_{10}$ | ws, gas_emcycle, aer_emcycle, CO, NO, $NO_2$,month, weekday, hour |

rh2: 2m relative humidity; slp: sea level pressure; t2: 2m air temperature; U: 10m U component of winds; V: 10m V component of winds; wd: 10m wind direction; ws: 10m wind speed; gas_emcycle: gasoline related emission cycle; aer_emcycle: diesel related emission cycle.

hyper-parameters: ntree (Number of trees to grow): 300 and mtry (Number of variables randomly sampled as candidates at each split): Rounded down square root of the number variables.

for each pollutant, and those performing the best for the testing dataset were selected. Table 3 presents the final set of predictive variables used in the RF model, as well as the hyper-parameters that were employed.

For the RF normalized, all variables were used and only the meteorological variables were normalized, following the approach described in Shi et al. (2021), which consists on resampling only the weather data over the whole study period, and is considered adequate for studying emission changes. We employed the `randomForest` package of the R programming language (Liaw and Wiener, 2002), and used the `rmweather` for the normalization process (Grange et al., 2018; Grange and Carslaw, 2019).

## 2.6  Random Forest model evaluation and assessment tools

The RF model was tested for adequate performance, focusing on the reproduction of: (i) the hourly concentrations, (ii) the mean diurnal cycles and (iii) the mean value. For each pollutant, the normalized mean bias (*NMB*) and the Pearson correlation

coefficient ($r$) for the hourly concentrations were calculated. The diurnal cycles were comparatively assessed by graphical inspection of the temporal series of the mean values and spreads of the modeled and observed concentrations of each pollutant.

$$NMB \, [\%] = \frac{\sum\limits_{k=1}^{N} (M_k - O_k)}{\sum\limits_{k=1}^{N} O_k} \times 100 \tag{1}$$

$$r = \frac{1}{N-1} \sum\limits_{k=1}^{N} \left( \frac{M_k - \bar{M}}{\sigma_M} \right) \left( \frac{O_k - \bar{O}}{\sigma_O} \right) \tag{2}$$

The *NMB* is useful for comparing pollutants that cover different concentration scales and it is defined as the difference between modeled and observed mean concentrations, normalized by dividing by the mean observed concentration for that period. The $r$ coefficient is useful to measure the linear relationship between two variables.

To detect, locate and characterize different pollution sources (Carslaw and Beevers, 2013; Grange et al., 2016), bivariate polar plots were built considering observations and RF results, using the openair library of the R programming language (Carslaw

and Ropkins, 2012; R Core Team, 2019). These plots provided a graphical support to analyze air pollutant concentrations together with wind speed and wind direction with and without COVID-19 restrictions. We also calculated them for March, April and May 2019 (MAM2019), so as to have a baseline to identify sources of the different pollutants.

Partial dependencies plots were also built using the rmweather library of R (Grange et al., 2018; Grange and Carslaw, 2019) to highlight the relationships between pollutant concentrations and all explanatory variables presented in Table 3, and can be

seen in the Supplementary Information (Figs. S9 to S11). By obtaining the prediction from the random forest model for each unique value of a specific explanatory variable, these plots allow us to analyze how this dependency varies for different values of the explanatory variable, and therefore help us to detect non-linear relationships, which are highly relevant in air quality.

## 3   Results and discussion

### 3.1   Analysis of the results of the Random Forest models

In general, modeled CO, NO, $NO_2$ and $PM_{10}$ concentrations in both sites were in good agreement with the corresponding observations (see Table 4), with NMB $< 6\%$ for both sites for the testing dataset. The Pearson correlation coefficient during testing ($r_t$) was above 0.7 for all pollutants, except $PM_{10}$. That was probably due to both a complex chemistry, with primary and secondary processes being highly relevant, and due to the effect of a few regional events during the period with a high effect on particulate matter. In addition, calculations of diurnal cycles utilizing RF outcomes reproduced adequately the clear

bimodal behavior of CO, NO, and $NO_2$ (Fig. 3). Nevertheless, biases during the BLD period are moderately larger than during the testing period. This is to be expected, given that the model was optimized to reproduce the testing period.

The results for $O_3$ were also satisfactory, particularly considering its secondary nature with complex dynamics, which depends on multiple factors such as radiation energies, VOC and $NO_x$ concentrations and their ratio (Seinfeld and Pandis,

**Table 4.** Summary of the evaluation statistics used in Random Forest predictive model for the testing dataset ($t$) and evaluation period (BLD).

| | | $NMB_{BLD}$ [%] | $NMB_t$ [%] | $r_t$ |
|------|----------------|------|------|------|
| PC | CO | 3.7 | 1.9 | 0.72 |
| | NO | -3.1 | 5.5 | 0.90 |
| | $NO_2$ | -5.1 | 3.0 | 0.78 |
| | $PM_{10}$ | -4.8 | 1.1 | 0.64 |
| CNEA | CO | 9.6 | 1.8 | 0.73 |
| | NO | -0.4 | 5.6 | 0.75 |
| | $NO_2$ | 3.9 | -0.3 | 0.91 |
| | $SO_2$ | 6.3 | -0.4 | 0.70 |
| | $O_3$ | 7.0 | 2.2 | 0.85 |

1998). Model performance indicators were $NMB_t$=2.2% and $r_t = 0.85$. Other processes involved in $O_3$ chemistry (like the
$O_3$/VOCs and $O_3$/$NO_x$ ratios) in the MABA were analyzed as a further way to test the RF model performance. The $O_3$–CO
ratio was used as a proxy for VOCs because direct VOCs observations were unavailable in the MABA and traffic-borne VOCs
are intimately linked to CO (Bon et al., 2011; Cazorla et al., 2020). Overall, above 75% of $O_3$–CO and $O_3$–$NO_x$ hourly ratios
from RF were within a factor of 2 of those resulting from the observations (Fig. S4 of the Supplementary Material). The
Pearson correlation coefficients between observed and estimated $O_3$–CO and $O_3$–$NO_x$ hourly ratios were found to be 0.85 and
0.9 respectively. In this context, this model was suitable to reproduce not only the levels of primary contaminants in the two
analyzed sites, but also the formation of $O_3$ at the CNEA site. The diurnal cycle of $SO_2$ (Fig. 3) during the BLD period had a
sharp peak between 18:00 and 20:00 that could not be entirely captured by the model, but it was linked to a day of particularly
high concentrations during that time period. Concentrations from 12:00 to 17:00 were also overestimated during the BLD.

Figure 3 shows that, during the BLD, the diurnal cycles of $O_3$ and $SO_2$ estimated using RFN are noticeable different
from those calculated using RF and the observations. This is further evidence that the atmospheric conditions can affect the
concentrations of pollutants in a relevant way under certain weather conditions.

One of the advantages of building a random forest model is that it could provide the key components that reflect the non-
linear relationship among the emissions, the chemistry, and the meteorology, by analyzing variables such as the permutation
difference (variable importance, Figs. 5 and S6 to S8 of the Supplementary Material) and the partial dependencies. The analysis
of the variable importance plots shows that the boundary layer height and the wind speed were important variables to predict
CO concentrations in both sites for normalized and not normalized models. This result is consistent with the fact that, at the
temporal scale studied here, CO can be considered as a passive tracer (Saide et al., 2011). For $NO_2$ and NO the most important
variables were the other pollutants included in the models and the surface temperature (Table 3), which was also expected,
because temperature has influence in $NO_x$ chemistry. In the case of $O_3$, the model was dominated by the concentrations of $NO_x$

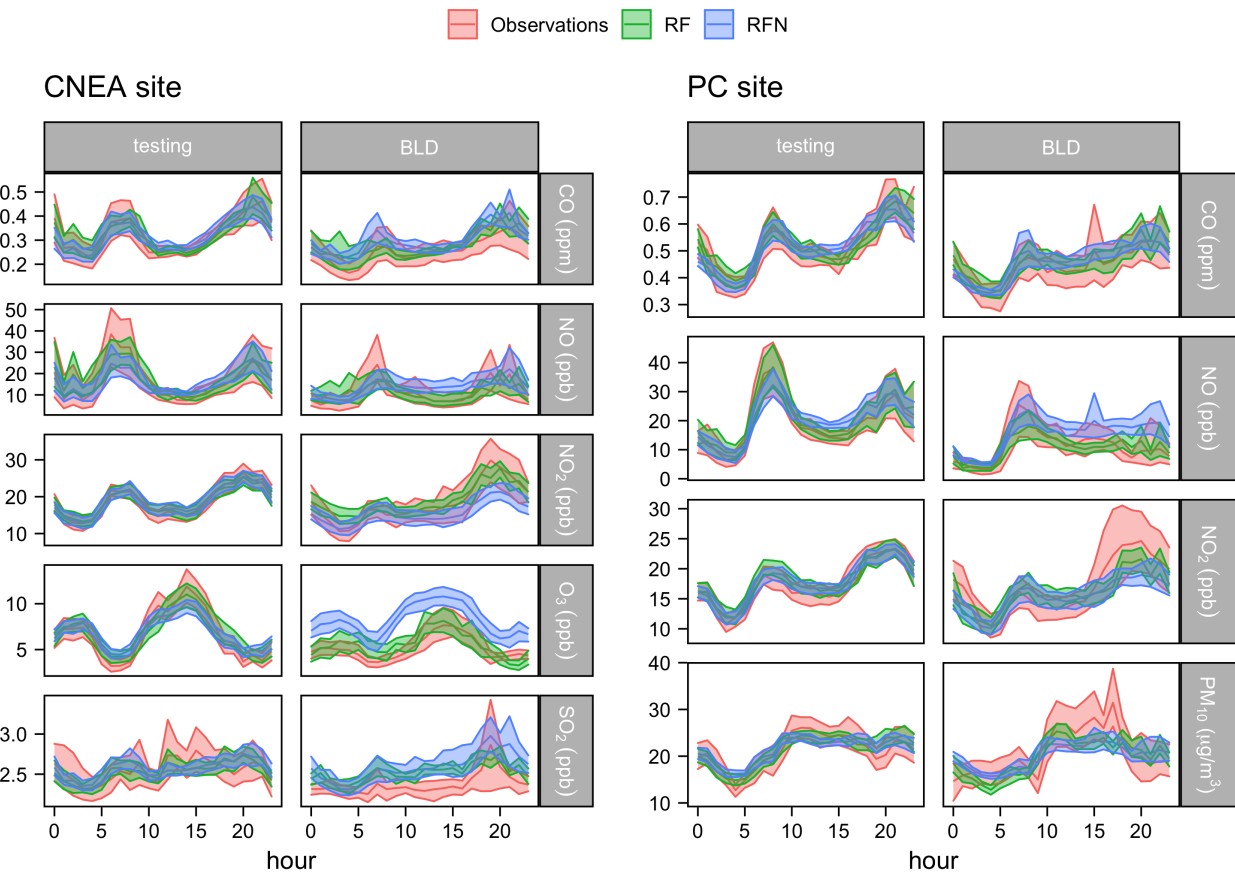

**Figure 3.** Mean diurnal cycles for the testing dataset and for the evaluation period (BLD). The line represents the average diurnal cycle and the shaded area represents the standard deviation.

and CO, with $NO_2$ being the most relevant, which is consistent with $O_3$ chemistry (see sections 3.2.2 and 3.2.3). The variable importance plot for $PM_{10}$ model show that CO and $NO_2$ are the most important variables to predict $PM_{10}$ concentrations. This was also expected because in Buenos Aires around 65 % of the $PM_{10}$ is $PM_{2.5}$ and the latter is highly correlated with CO (Arkouli et al., 2010).

Partial dependencies plots (Figs. S9 to S11 of the Supplementary Material) enlighten the relationships between pollutant concentrations and temperature. As an example, in CNEA, while CO, NO and $NO_2$ concentrations were inversely related with temperature, $SO_2$ presented the opposite behavior. As described by Grange and Carslaw (2019) this relationship of $SO_2$ with temperature could be associated with shipping emissions. This is also consistent with the fact that there is also a high partial dependence with wind directions from 0 to 100° (Figs. S3 and S11 of the Supplementary Material), which is the range of winds that bring air masses from La Plata River.

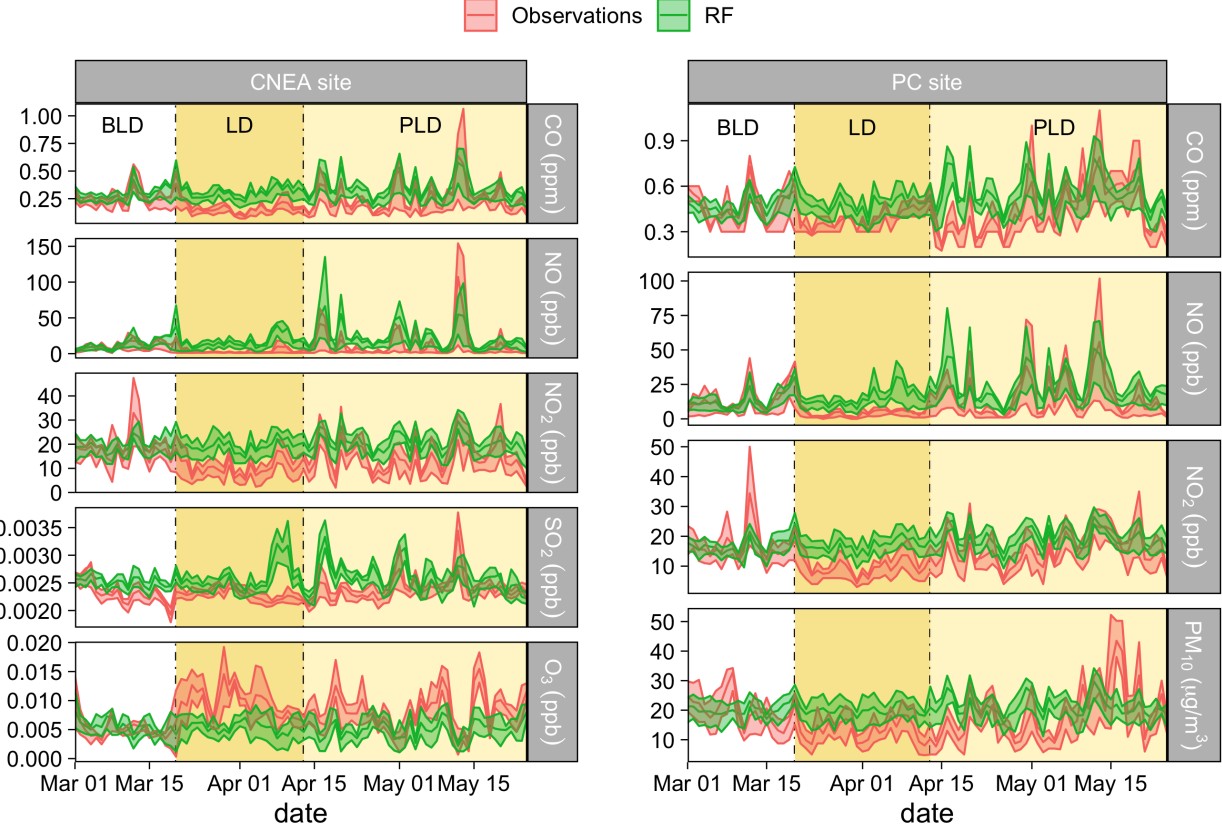

**Figure 4.** Average daily concentrations for CNEA and PC sites. The line represents the 24 h average concentration and the shaded area represents the daily levels between the 25th and 75th percentile.

## 3.2 Quantifying and analyzing the changes in concentrations during the lockdown periods

We discuss here the relative changes of: (1) measured concentrations during the LD and the PLD periods in comparison with RF outputs for the same period, and (2) normalized measured concentrations during the LD and the PLD with normalized concentrations during the same periods, but for 2019 (March 20th to April 12th, and April 13th to May 25th). The corresponding percent relative changes ($RC_{RF}$ and $RC_{RFN}$) were estimated using the expressions presented in Eqs. 3 and 4. We make use of $RC_{RF}$ to quantify the amount of change with respect to a BAU scenario for the particular meteorological conditions that happened during the two lockdown periods, and $RC_{RFN}$ to quantify the effects of the changes in emissions of these pollutants sources, rather than meteorological or environmental effects of particular atmospheric conditions.

$$RC_{RF}[\%] = \frac{\overline{Obs}_{LD,PLD} - \overline{RF}_{LD,PLD}}{\overline{RF}_{LD,PLD}} \times 100 \qquad (3)$$

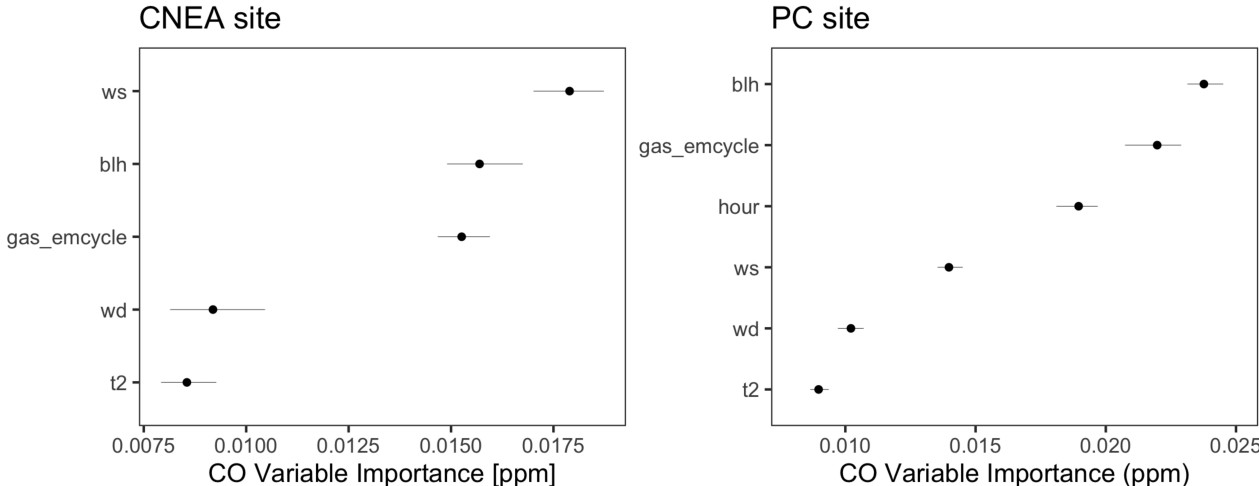

**Figure 5.** Variable Importance plot (permutation difference) for CO variable (ppm) in CNEA (left) and PC (right) for RF model.

$$\mathrm{RC_{RFN}}[\%] = \frac{\overline{\mathrm{RFN}}_{LD,PLD} - \overline{\mathrm{RFN}}_{\text{same periods 2019}}}{\overline{\mathrm{RFN}}_{\text{same periods 2019}}} \times 100 \tag{4}$$

where $\overline{\mathrm{Obs}}_{LD,PLD}$ corresponds to the hourly mean concentrations observed during the LD or the PLD, and $\overline{\mathrm{RF}}_{LD,PLD}$ is the corresponding predictive RF for the same periods, $\overline{\mathrm{RFN}}_{LD,PLD}$ refers to the data for the LD and the PLD with the normalization of the meteorological variables, which was compared with the meteorologically normalized data of the same periods in 2019.

Being both monitoring sites highly influenced by vehicular emissions, the traffic reduction of $\sim 80\%$ that was registered
during the LD period led to a significant air quality improvement of primary pollutants (Fig. 6). In almost all cases, except for CO in PC and for $SO_2$, the meteorological conditions amplified the change, as shown by the fact that $\mathrm{RC_{RFN}}$ is smaller than $\mathrm{RC_{RF}}$. This is consistent with the results obtained by Shi et al. (2021).

On the other hand, observed $O_3$ levels were 80% and 57% higher in comparison with the RF estimations for the LD and the PLD respectively. However, the fact that this increment was considerably smaller when the meteorology was normalized
indicates that this change was strongly enhanced by the meteorological conditions that occurred during that period.

Figure 4 displays the differences in daily concentrations between observations and RF estimates for the three considered periods (BLD, LD and PLD). For CNEA, CO and $NO_x$ observations and predictions for the BLD period showed *NMB* <10%. Noticeably, most of the changes were observed right from the day after lockdown. Pollutant levels were almost fully recovered by the last week of the PLD period.
In what follows, the results are presented by species, highlighting the most relevant relative changes in concentrations. The results of the meteorological normalization are used to evaluate the effects of the changes in emissions of particular pollutants,

**Table 5.** Summary of the average concentrations for the BLD, the LD and the PLD and the relative changes for the LD and the PLD for PC and CNEA sites estimated by RF and RFN. In every case, the RC were calculated considering the mean value for each period.

| | BLD | | | LD | | | | | PLD | | | | |
| --- | --- | --- | --- | --- | --- | --- | --- | --- | --- | --- | --- | --- | --- |
| | Concentrations | | | Concentrations | | | RC [%] | | Concentrations | | | RC [%] | |
| | obs | RF | RFN | obs | RF | RFN | RF | RFN | obs | RF | RFN | RF | RFN |
| PC | | | | | | | | | | | | | |
| CO (ppm) | 0.43 | 0.46 | 0.46 | 0.39 | 0.49 | 0.44 | -20 | -20 | 0.45 | 0.56 | 0.51 | -19 | -7 |
| NO (ppb) | 11.6 | 10.9 | 15.6 | 5.2 | 15.6 | 14.6 | -67 | -35 | 15.2 | 24.3 | 19.0 | -37 | -15 |
| NO$_2$ (ppb) | 16.8 | 16.0 | 15.3 | 9.8 | 17.3 | 13.5 | -43 | -28 | 15.5 | 20.0 | 16.4 | -20 | -13 |
| PM$_{10}$ ($\mu$gm$^{-3}$) | 20.5 | 19.6 | 20.1 | 13.6 | 20.2 | 17.9 | -33 | -20 | 18.4 | 21.4 | 20.6 | -14 | -7 |
| CNEA | | | | | | | | | | | | | |
| CO (ppm) | 0.26 | 0.28 | 0.3 | 0.17 | 0.31 | 0.26 | -45 | -26 | 0.25 | 0.34 | 0.31 | -26 | -11 |
| NO (ppb) | 11.4 | 11.4 | 14.1 | 4.3 | 17.4 | 10.3 | -75 | -47 | 14.6 | 23.1 | 15.0 | -37 | -21 |
| NO$_2$ (ppb) | 18.3 | 19.1 | 15.6 | 10.6 | 19.5 | 10.7 | -46 | -36 | 14.4 | 20.7 | 14.0 | -30 | -15 |
| SO$_2$ (ppb) | 2.4 | 2.5 | 2.64 | 2.3 | 2.6 | 2.4 | -12 | -20 | 2.4 | 2.6 | 2.5 | -8 | -15 |
| O$_3$ (ppb) | 5 | 5.5 | 9 | 9.6 | 8.1 | 10.7 | 80 | 27 | 8.0 | 5.1 | 8.8 | 57 | 5 |

as a consequence of the restrictions previously discussed. Bivariate polar plots were used to distinguish potential sources that impact the monitoring sites (Figures 8 and 9).

### 3.2.1 Carbon monoxide

As shown in table 5 and discussed below, there was a reduction in CO levels when the highest restrictions were in place (LD). However, the behavior of this pollutant when the restrictions were partially lifted (PLD) differed depending on the measuring site.

In PC, the recovery of traffic during the PLD ($RC_{RF}^{PLD} = -19\%$) did not result in a smaller relative change with respect to a scenario with higher restrictions ($RC_{RF}^{LD} = -20\%$). Nevertheless, as shown by $RC_{RFN}$, decoupling the effects of the

330 meteorology, the relative change was -20% in the LD, but only -7% in the PLD, with respect to the normalized values for the same periods in 2019. These results show the influence that the particular meteorological conditions had on CO concentrations in PC. On the other hand, in CNEA, the partial lift of restrictions during the PLD resulted in a smaller relative change in CO concentrations that is clear both for the particular meteorological conditions of the two periods (-45% for $RC_{RF}^{LD}$ vs -26% for $RC_{RF}^{PLD}$) and for the normalized model (-26% for $RC_{RFN}^{LD}$ vs -11% for $RC_{RFN}^{PLD}$).

The observed CO had lower concentration values and flatter diurnal patterns than our simulations of a BAU scenario (Fig. 6). This reduction far surpasses any bias detected in RF simulations, particularly during rush hours, where RF showed close to no bias (Fig. 3). This is particularly true in CNEA, where the general reduction of CO was larger. For this pollutant, there are no

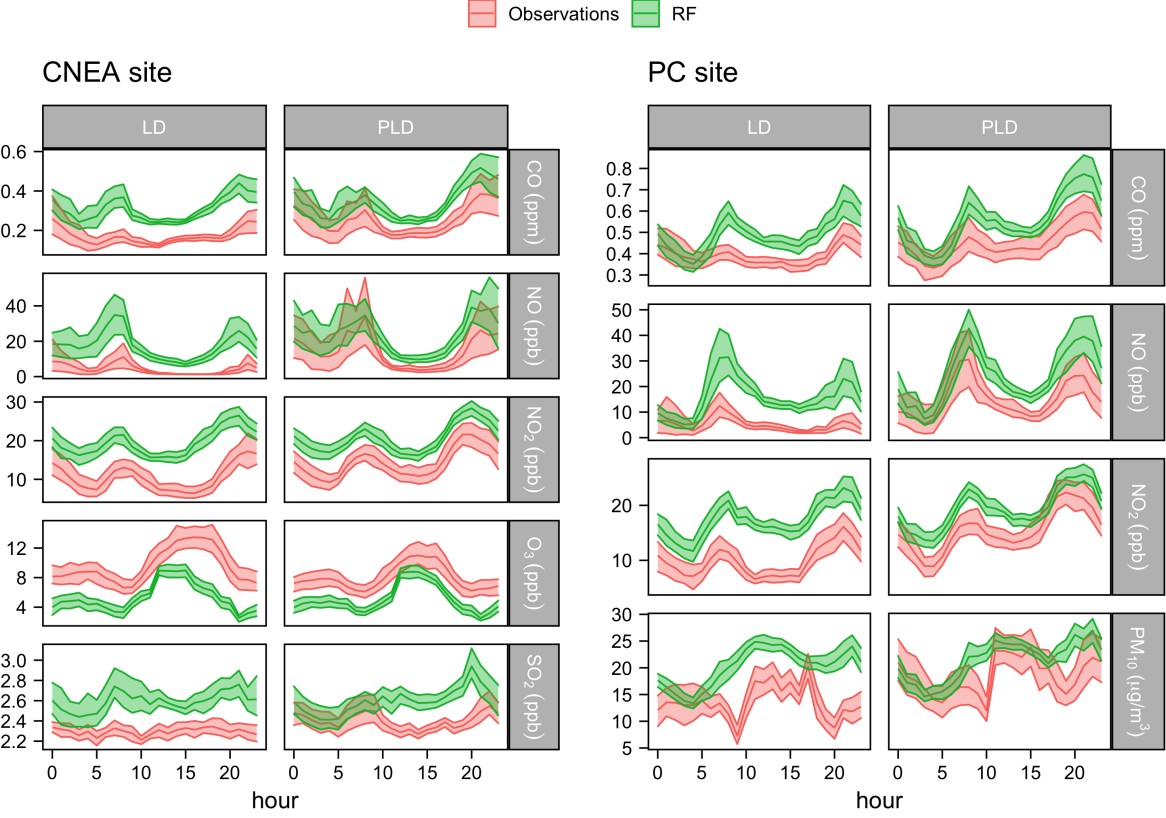

**Figure 6.** Mean diurnal cycle for the different pollutants for the LD (from 20 March 2020 to 13 April 2020) and the PLD (13 April 2020 to 25 May 2020) for both sites. The line represents the average diurnal cycle and the shaded area represents the standard deviation.

big differences between the changes in the normalized diurnal cycle and those obtained comparing the RF predictive model with the observations (Figs. 6 and 7).

As shown in Figure 8, for the CNEA site during MAM2019, concentrations were similar for all wind directions and speeds (up to 8 m/s). The largest relative changes between the 2020 observations and the RF simulations were when winds were coming from the E and SE (both for the LD and the PLD). These were probably due to a reduction in traffic on the highway (see Section 2.4.1), which according to Diaz Resquin et al. (2018), is one of the principal sources of fuel combustion emissions.

An equivalent analysis for PC (Figure 9) yielded similar results during MAM2019, although concentrations seemed to be largest when winds were from the W. However, relative changes during the LD and the PLD did not seem to have a clear dominant wind direction. During the PLD, sources from the W reappeared.

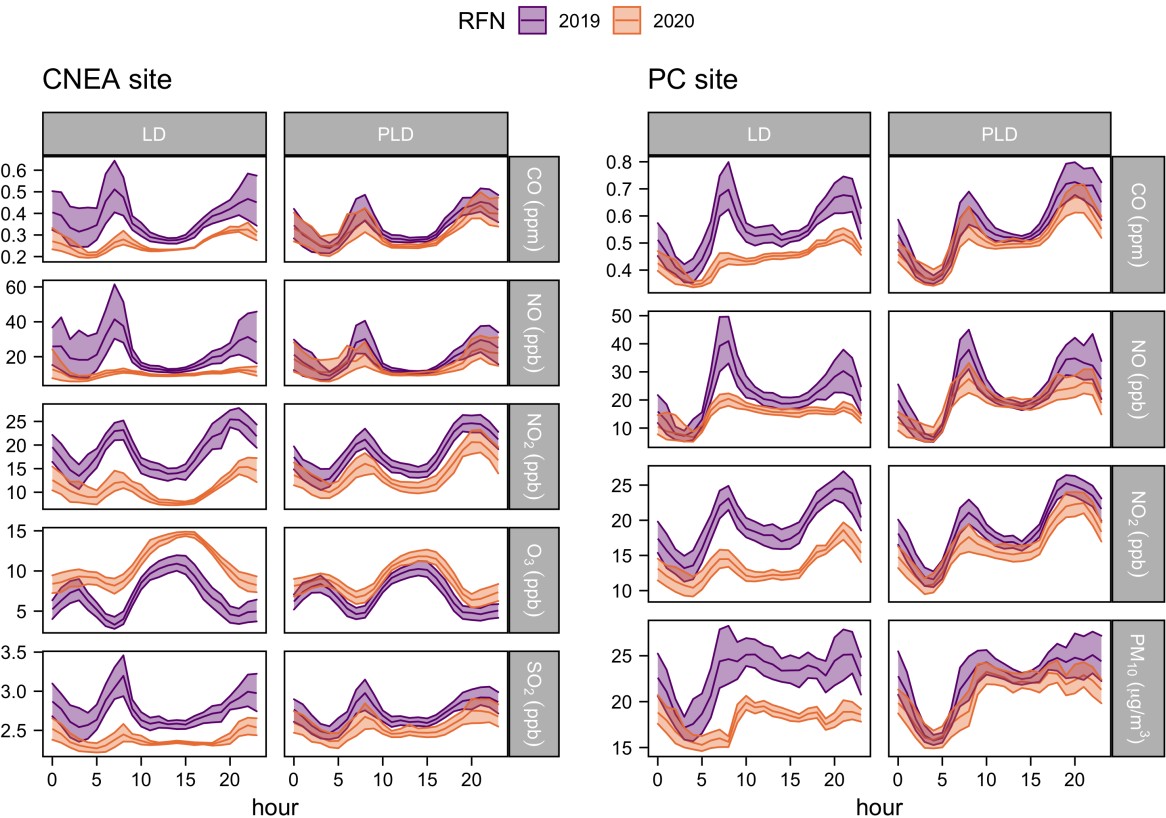

**Figure 7.** Mean diurnal cycle for different pollutants for the LD, PLD and MAM2019 periods, with meteorological normalization, for both sites.

### 3.2.2 Nitrogen oxides

The drastic reduction of vehicular emissions impacted positively in the NO and $NO_2$ levels. As shown in Table 5, during the LD period, NO levels were one third and one fourth of the estimated value for a BAU scenario in PC and CNEA respectively.

The relative change for $NO_2$ was ∼-45%. During the PLD, the relative change was smaller: -37% for both sites for NO, -20% and -30% for $NO_2$ in PC and CNEA respectively.

In both sites, the relative changes of nitrogen oxides were larger than those of CO. Arguably, this indicates that the power plants did not contribute in any major way to the observed differences. This is probably due to a reduced circulation of diesel vehicles, which are the major nitrogen oxides emitters (D'Angiola et al., 2010; Ghaffarpasand et al., 2020).

$RC_{RFN}$ shows that these changes were consistently enhanced by the meteorological conditions during that period, so that the changes with a meteorological normalization are between two thirds and half as large as those without.

We also see a flattening of the diurnal cycles of NO during the LD, both in the RF predictive model and in the analysis with normalized meteorology (Figures 6 and 7). The bimodal curve is partially recovered during the PLD. This indicates, once

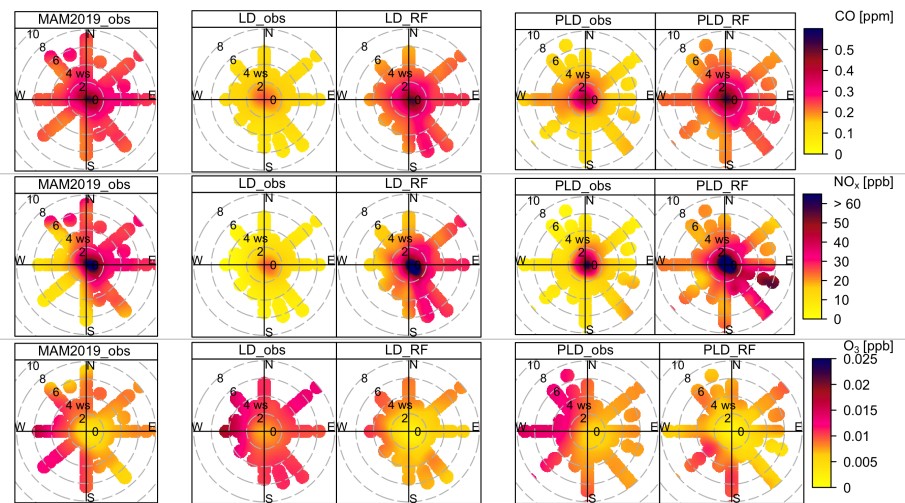

**Figure 8.** Bivariate polar plot for CNEA of hourly means for observations during MAM2019 and lockdown periods versus the BAU scenario estimated with RF model. The radial axis represents wind speed, the angular axis represents wind direction, and the color scale represents pollutant concentrations.

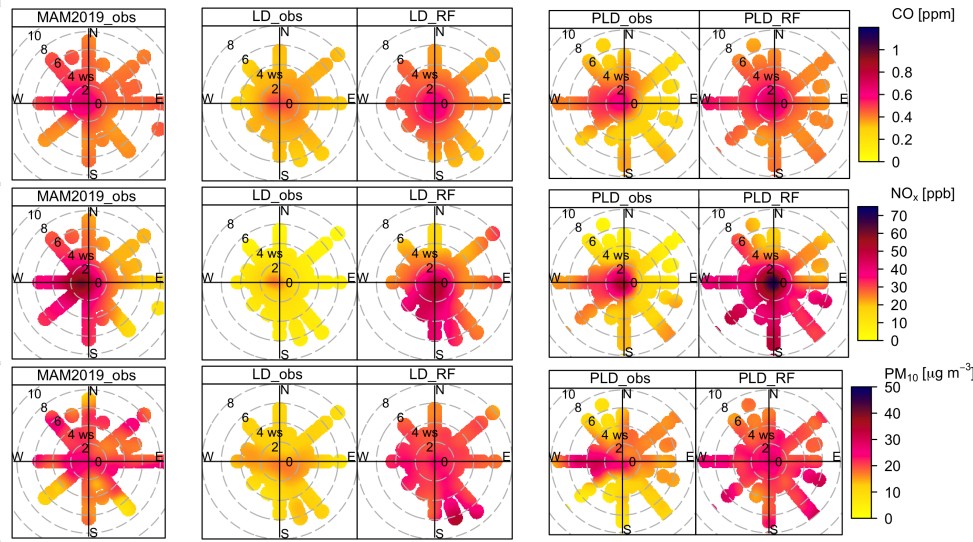

**Figure 9.** Bivariate polar plot for PC of hourly means for observations during MAM2019 and lockdown periods versus the BAU scenario estimated with the RF model. The radial axis represents wind speed, the angular axis represents wind direction, and the color scale represents pollutant concentrations.

again, the strong role of traffic emissions in NO concentrations. $NO_2$, however, preserves most of its bimodal nature, albeit
somewhat diminished. Although a clear explanation for this fact is hard to find, while NO is predominantly a primary pollutant,

NO$_2$ is partially secondary in origin, and is largely influenced by NO, O$_3$ and HO$_x$ concentrations, as well as radiation and other meteorological parameters (Han et al., 2011; Brasseur and Jacob, 2017). NO is photochemically converted to NO$_2$ by reacting with O$_3$ during the morning, but is converted back to NO due to photolysis during the daytime, generating an O radical that regenerates O$_3$. At night, O$_3$ and NO$_2$ react with each other, in a chain of reactions that end up generating HNO$_3$ in the aqueous phase of aerosols. The diurnal cycle of this photochemical processes should be largely regulated by the solar radiation, and therefore unaffected by the restrictions. This remains true even if NO emissions are flattened and the total concentrations of NO$_2$ are also clearly lower, particularly during daytime.

Fig. 8 shows the bivariate polar plots of the NO$_x$ concentrations at the CNEA site. The bivariate polar plot in MAM2019 provides evidence for two main contributing sources. One source was due to air masses from E-SE directions at low wind speeds and the second source was associated with higher wind speeds from N-NW direction. The source to the E-SE could be dominated by ground-level road traffic emissions that are closer to the site because high concentrations under low wind speeds are indicative of surface emissions released with little or no buoyancy (Uria-Tellaetxe and Carslaw, 2014). Also, the wind direction where this source was dominant corresponds to the highway previously described in Section 2.4.1. The source to the N-NW was associated with high concentrations at high wind speeds, which is indicative of emissions at a greater distance. It is plausible to attribute these NO$_x$ levels to the main access avenue that connects the city with the suburbs and is located in this direction, due to the presence of heavy-duty diesel vehicles and buses and the number of flowing traffic stops. During the LD and the PLD, the highest RC$_{RF}$ were present when winds were coming from the highway. This serves as further evidence that the observed effects were mainly due to changes in traffic, and not to the changes in residential emission patterns due to lifestyle changes during the lockdown.

In the case of PC, as shown in Figure 9, during MAM2019, the main sources seemed to be located to the W and SW of the station. These two directions entailed the largest changes due to restrictions during the LD period. During the PLD period, in a similar manner than CO, the sources to the W were partially restored (although concentrations from the SW remained low).

### 3.2.3 Ozone

By contrast to the other pollutants considered, the O$_3$ was higher when compared to a no-restrictions scenario. Its relative changes estimated using the RF predictive model were 80% and 57% during the LD and the PLD periods respectively.

Recent studies of the lockdown effects on atmospheric composition have also reported large O$_3$ increases at urban sites and indicated the need of analyzing changes in precursor emissions and meteorological parameters in light of their role in the nonlinear response in the O$_3$ concentrations (Ordóñez et al., 2020; Tobías et al., 2020; Nakada Kondo and Urban, 2020; Shi and Brasseur, 2020). Hence, consideration of the joint effects of the changes on precursors and meteorology are of great value to understand the differences between the relative changes estimated using RF concentrations. Based on Figures 4 and 6, we provide plausible explanations for these discrepancies.

It is well known that decreasing nitrogen oxides levels in a VOC-limited regime tend to increase O$_3$. It is most likely that the lower concentrations of freshly emitted NO registered during the LD and the PLD in CNEA provoked a decline in the local scavenging of O$_3$, leading to higher O$_3$ concentrations, particularly in the morning (Tobías et al., 2020; Nakada and Urban,

2020). Even though NO is the pollutant that had the highest relative decrease during the LD and the PLD, its reduction is not enough to explain the overall relative increase in $O_3$, and therefore $NO_2$ might have played a role as well. Lower $NO_2$ levels could have also resulted in more OH to initiate $O_3$ production because the inhibition of termination reaction favors faster $O_3$ accumulation (Seguel et al., 2012).

With respect to the role of aerosols in $O_3$ formation, it is worth noting that a significant decrease in $PM_{10}$ was registered
in PC. This likely implied consequent reduction not only in the mass concentrations of $PM_{2.5}$ and $PM_1$, but especially in the number concentration of fine and ultrafine particles (Arkouli et al., 2010; Gelman Constantin et al., 2021). A similar situation most likely occurred in CNEA. This could have led to greater photolysis due to the decrease in emissions of fine particles as a consequence of the vehicular restrictions imposed during the lockdowns, which in turn could have led to higher $O_3$ concentrations (Wang et al., 2019).

In this case, meteorological factors were clearly highly relevant, as can be seen by the fact that the relative change estimated with the RFN model is far smaller (27% for $RC_{RFN}^{LD}$ and only 5% for $RC_{RFN}^{PLD}$). The effects of meteorology can be rather complex since the $O_3$ precursor concentrations and reaction rates are affected in multiple ways (Wang et al., 2017). Although meteorological variables such as the temperature and relative humidity are highly relevant for ozone production and chemistry, they were tested as explanatory variables and, in this case, led to model degradation. However, we submit that their effects are
indirectly taken into account by the chemical species that were employed (CO, NO, $NO_2$ and $SO_2$). Solar radiation, which is highly relevant for $O_3$ chemistry, is also linked to the variable daynight. In this particular case, during the LD, elevated $O_3$ concentrations occurred on days with high temperatures and low winds, which favor the photochemical production of $O_3$ and the accumulation of ozone and its precursors.

When the meteorology is normalized, the valleys at 7:00 and 20:00 are clearly less marked during 2020 than during 2019,
and almost disappeared during the LD compared with the normalized values for the same period of the previous year (Fig. 7). This is probably due to the lower concentrations of nitrogen oxides, that therefore are less efficient at titrating $O_3$ (Brasseur and Jacob, 2017).

As expected, the bivariate polar plots (Figure 8) show that $O_3$ behaved opposite to $NO_x$, having the largest increases when winds came from the E and SE during the LD and also when they came from the E and NW during the PLD.
From these results, we can also derive that the area where the CNEA site is located behaves as a region with a VOC-limited chemical regime, because the reduction in $NO_x$ emissions caused an increase in ozone concentrations (Blanchard and Fairley, 2001; Heuss et al., 2003; Yarwood et al., 2003; Blanchard and Tanenbaum, 2006). We identified a similar behavior of increasing $O_3$ concentrations under decreasing $NO_x$ levels when analyzing the 2019 data for weekends (Fig. S5 of the Supplementary Material). This is related to the denominated weekend effect in a VOC-limited regime (Koo et al., 2012).

**3.2.4 Sulphur dioxide**

During the LD, the $SO_2$ concentrations were slightly lower than those of the simulated BAU scenario ($RC_{RF}$ of -12%). Although this change is not as large as in the other species for the particular meteorological conditions that occurred during the

period, if we consider a normalized meteorology, we observe a relative change of -20%, which is about as large as the change observed in, for example, CO. There was smaller relative change during the PLD, which was similar for RF and RFN.

While all other species in this study are mostly controlled, directly or indirectly, by on road traffic emissions, according to our findings $SO_2$ concentrations are largely influenced by shipping emissions (see section 3.1). This might be the reason why $SO_2$ is the specie with a larger change after normalizing the meteorology.

Another possible reason for having a smaller relative change in $SO_2$ concentrations is that the vehicle emissions of heavy-duty diesel trucks are another relevant source in Buenos Aires. These are mainly associated with essential activities, and might have not been affected as much by the restrictions. However, the partial flattening of the normalized diurnal cycle (Fig. 7) is still probably related to changes in this particular sort of traffic.

### 3.2.5 Particulate matter 10 μm

During the LD, $PM_{10}$ had a relative change of -33% compared to what would be expected for that specific period under previous emissions. This effect was once again enhanced by the meteorological factors, considering that $RC_{RFN}$ was only -20%. During the PLD, similarly to what happened with other pollutants, the concentrations had a relative change only about half as large (-14% for the RF predictive model, and -7% for the RFN).

When winds are taken into account (Figure 9), we observe a general reduction from all directions during the LD. Two sources account for this: (i) the anthropogenic $PM_{10}$ emissions close to the monitoring site that were mostly from vehicle diesel combustion and soot resuspension and (ii) natural sources, such as dust emissions, from the nearest large open area. In a similar fashion to CO and $NO_x$, sources from the W were reestablished during the PLD.

### 3.3 Vehicle emission reduction strategies and air pollution in the MABA

Although, as expected, most pollutants were noticeably reduced during the LD due to the restrictions imposed, $O_3$ was an exception. Strategies for controlling pollution from vehicular emissions in the MABA must take into account the relative reductions of $NO_x$ and VOCs to avoid an unintended increment in $O_3$ concentrations. The atmosphere in the MABA is usually cleaned up during the night, due to a flat topography and the city's wind dynamics. Therefore, criteria pollutants rarely surpass air quality norms. Even though no specific policies to reduce them have been implemented, recently announced greenhouse gas emission mitigation policies affecting on-road mobile emissions may have a major impact. These include (i) technological advances in diesel buses, that should reduce $NO_x$ and $PM_{10}$, without a major impact in VOCs and (ii) an increase of the fraction of electric cars, which should reduce $NO_x$ and VOC concentrations. Thus, if $NO_x$ emissions decrease like they did during the COVID lockdown, this will likely result in an increment in tropospheric $O_3$ in the MABA if no additional measures regarding VOCs emissions are included, which could be of particular importance for some weather conditions. In fact, under the VOC-limited regime identified for the MABA, control of VOCs emission would be more efficient to reduce local peaks in $O_3$.

This highlights the importance of having comprehensive air quality policies rather than focusing on reductions in individual pollutants.

## 4 Code and data availability

Hourly concentrations of CO, NO, $NO_2$, $SO_2$ and $O_3$ in CNEA, are available in .csv format at https://data.mendeley.com/datasets/h9y4hb8sf8/1 (Diaz Resquin et al., 2021). We also provide an introductory R notebook with some baseline simulations for the predictive model. For PC regulatory averages are publicly available and can be accessed through their website (https://data.buenosaires.gob.ar/dataset/calidad-aire). Nevertheless hourly data is not regularly reported, but can be requested to the Environmental Protection Agency of Buenos Aires City. To enable a machine learning quick start to reproduce the baseline experiments, we also added to the dataset the meteorological data used to run the simulations. It is publicly available at the website of the National Weather Service (https://www.smn.gob.ar/descarga-de-datos).

## 5 Summary and conclusions

In this study, we present novel air quality data for a residential site located in the Metropolitan area of Buenos Aires that includes concentrations of CO, NO, $NO_2$, and, of particular importance for the city, $SO_2$ and $O_3$. One year of these data, together with data from a public monitoring station, were used to train Random Forest models. The performance of the models was tested on the basis of observations registered both with a separate testing set during the training period and with data before the outbreak of the COVID-19 pandemic. Observations in the two first phases of the lockdown measures imposed were compared with business-as-usual RF concentrations to assess the change with respect to the air pollutant concentrations that would have occurred without the lockdown. Simultaneously, a meteorological normalization using Random Forest was performed (RFN), and the normalized concentrations during these lockdown phases were compared with the normalized concentrations for the same periods during 2019. The main conclusions are listed below:

(i) The resulting set of explanatory variables for the different pollutants in each site provides evidence of the need for careful variable identification during the training period. Although ideally the best explanatory variables could be identified by trial and error by non-experienced users of random forest models with the support of variable importance plots, it is advisable to count with expert judgment for a meaningful and relatively fast selection.

(ii) The RF model was able to reproduce air quality observations at two monitoring stations in the MABA when evaluated for a 15-day period previous to the outbreak of the COVID-19 pandemic. This approach allowed predicting pollutant hourly mean values with a mean bias of less than 10% by using data of air quality, emissions and meteorology and analyzing the effect of wind direction and speed in pollutant concentration, which is useful when characterizing pollution sources.

(iii) During the lockdown, all primary pollutants had lower concentrations than what the RF framework would predict for a business-as-usual scenario. The relative change ranged from -12% ($SO_2$) to -75% (NO in the monitoring site of CNEA). In the case of all pollutants but $SO_2$, the relative changes were enhanced by the meteorology, as shown by the fact that, in absolute terms, $RC_{RF}$ was generally larger than $RC_{RFN}$. This difference was particularly large for $O_3$, probably due to its secondary nature and its complex chemical and photochemical production and destruction mechanisms. The

exception observed in the case of $SO_2$ is likely due to the importance of the wind direction, due to the relevance of the shipping emissions. The relative changes in pollutant concentrations are closely linked to both the traffic and the particular meteorological conditions. The use of bivariate polar plots is also helpful for identifying potential sources, while remaining relatively easy to implement.

(iv) RF estimations can be implemented at a low computational cost, and can be used to assess the changes that occurred in a specific period if an anomalous situation happened. It can also be used to forecast air quality conditions in the short term at a lower cost than CTMs, which could be of use for local authorities, considering that the MABA has thus far only six long-term air quality monitoring stations. When, as in this case, detailed temporal information on different emission sources is lacking (for example, traffic information from on-road sensors), it is essential to use a set of data in which the emissions are similar to those that are expected to be simulated. The model also allows to analyze the relations between different pollutants, which is of particular interest for those that have very complex chemistry, such as $O_3$. The observational input data needed for future RF simulations can be readily updated. The modeling framework developed in this study is user friendly, rather straightforward to implement and do not require large computational capacity. The methodology is amenable to be adapted to different time periods and sites and implemented by the technical staff of regulatory agencies. Expert advice may be needed during the selection of the predictive variables and model optimization.

(v) For assessing the effectiveness of a particular measure in AQ independently of particular meteorological conditions of specific periods, a meteorological normalization technique based on random forest can be used. This approach is relatively simple to implement with already existing R packages.

(vi) Although previous studies employed both techniques with similar aims, we postulate that the use of the RF predictive model and the meteorological normalization serve different purposes, and should be used accordingly. The predictive model can be used to analyze the changes in for particular weather conditions or, combined with a meteorological forecast, to forecast pollutant concentrations. On the other hand, the meteorological normalization makes it possible to evaluate the general impact in concentrations due to changes in emissions, decoupling the effects of particular meteorological conditions from the short-term emission changes from the AQ datasets.

(vii) In this work we provide the first year-long in situ observational dataset on tropospheric $O_3$ and $SO_2$, outside of an industrial area for the MABA in the last decade. We also provide co-located concentrations of CO, NO and $NO_2$.

(viii) According to our measurements, the MABA seems to be in a VOC-limited regime. If VOC emissions are not carefully regulated, a $NO_x$ reduction would imply an increase in the tropospheric $O_3$. Knowing how the concentrations of $O_3$ in the troposphere respond to reducing the emissions of its precursors is relevant when planning appropriate strategies to reduce CO, NMVOCs and $NO_x$ emissions. Even though this classification is limited due to the fact that we only have single point measurements, this could be a useful starting point for a more thorough characterization of the ozone regime in this urban area.

*Author contributions.* Díaz Resquín, Melisa: Conceptualization, Methodology, Validation, Data curation, Data Analysis, Formal analysis, Writing - original draft, review & editing. Lichtig, Pablo: Data Analysis, Formal analysis, Writing - original draft, review & editing. Alessandrello, Diego: Data Acquisition, Data curation, Writing - original draft. De Oto, Marcelo: Data Acquisition, Writing - original draft, review & editing. Castesana, Paula: Data Analysis, Writing - review. Rössler, Cristina: Data Analysis, Writing - original draft. Gómez, Darío: Funding acquisition, Supervision, Formal analysis, Writing - original draft, review & editing. Dawidowski, Laura: Funding acquisition, Conceptualization, Supervision, Formal analysis, Writing - original draft, review & editing.

*Competing interests.* The authors declare that no competing interests are present.

*Acknowledgements.* We want acknowledge the participation of the entire group of Atmospheric Chemistry of the National Atomic Energy Commission of Argentina (CNEA) to hold the campaign even during Lockdown. This study was carried out in part with the aid of grants PICT-O 2016-4802 (Agencia Nacional de Promoción Científica y Tecnológica, Argentina) and PICT 2016-3590 (Fondo para la Investigación Científica y Tecnológica). The research work that leads to the results has also been funded by the EU Horizon 2020 Marie Skłodowska-Curie project PAPILA (GA 777544, MSCA action for research and innovation staff exchange). The authors wish to thank the Environmental Protection Agency of Buenos Aires (APRA) and the National Weather Service of Argentina (SMN) for sharing air quality and meteorological data for this study. We also appreciate the editor and reviewers for their comments and recommendations that helped to improve the manuscript.

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
