# Peer review of "A machine learning approach to address air quality changes during the COVID-19 lockdown in Buenos Aires, Argentina"

_Earth System Science Data, 2021_

## Author Comment (AC1)

Answer Referee #1
**General comments to both referees:**
Most of the suggestions received have been adopted. From this, major changes have been made on the study, comprising: (1) the inclusion of Gini importance plots; (2) a modification of the set of predictive variables, adding the boundary layer height and the total cloud cover and (3) the normalization of the meteorological variables. On this basis, we build a new model to decouple the effect of the meteorology for the analysis of the relative changes during COVID-19 period. Although these suggestions led to an overall better implementation of the Random Forest model and a better subsequent analysis, the conclusions remain almost unchanged. Nevertheless, with the new estimated values for the relative changes, we will include substantive modifications in section 3.1.

This study was majorly inspired by Grange et al 2019 to use a random forest model (RF) to train, validate and predict the air quality concentration in a megacity of Argentina. Although the methodology is not new, this study has the potential to have a significant improvement to fill the data gap given the reason that some monitored trace gas concentrations were lacking but the pollutants are becoming a concern for local authorities. (line 145- 150) I would like to propose a few major revisions for the authors of this manuscript. After done with that improvement, I think it would be strong enough to publish on earth system science data:

| | |
|---|---|
| 1. Random forest models indeed are easy and quick to train. But if this study is only focusing on predicting the time series of pollution during COVID-19, there are better and more efficient machine learning models such as ARIMA compared with random forest. The key reason for many studies that authors mentioned in this article used RF is because it could provide key components reflecting the non-linear relationship among emissions, chemical reaction, and meteorological effects. Please see figure 3 in Grange et al 2019 and figure 1 in Yang et al. It is easy to generate that Gini importance either through python or R. Since this code the authors used were based on R, I would suggest they follow the | Following your suggestion, we generated the Gini importance analysis using the *ranger* package in R for all the variables considered in the model. As you mentioned, these results were helpful for us to understand the underlying role (non-linear effects) of some variables in the concentrations of the pollutants analyzed. As an example, Gini plots reveal the importance to consider the boundary layer height as an explanatory variable, particularly for CO and NO description. On the basis of this analysis, we modified the set of the explanatory variables chosen for each model, adding the boundary layer height (blh). For example, for the CNEA site: (1) for CO the set of explanatory variables was change from {t2; rh2; U; V; gasoline diurnal cycle} to { t2; rh2; sea level pressure; ws; wd; blh; total cloud cover; gasoline diurnal cycle; hour} and (2) for NO the set was modified from {t2; rh2; U; V; gasoline and diesel diurnal cycles} to {t2; rh2; sea level pressure; ws; wd; blh; total cloud cover; gasoline and diesel diurnal cycle; CO;hour; month, weekday}

Regarding the aim of the study, note that we presented 3 different goals. Two of them (lines 65-68) are: (1) the prediction of the time series during COVID-19 and (2) the development of a model for air quality forecasts for the Metropolitan Area of Buenos Aires at a low computational cost. In line 74 and 75, we also explain that we are |

| | | |
|---|---|---|
| | code from Grange et al 2019 to generate the Gini importance plot. After getting those plots, you can compare your RF results with the reasons from previous literature to see if it makes sense. | providing (3) the first $O_3$ and $SO_2$ observational datasets in more than a decade in Buenos Aires. However, from your comment, we realized that (2) and (3) were not highlighted in the abstract. Therefore, we will include some modifications in the abstract. |
| 2. | The authors also mentioned several times the meteorological impacts on air pollution. The earliest signal has been seen during COVID was the study in China from Le et al, 2020 which authors may consider mentioning this study result. Then it would also be beneficial to consider using RF models to generate new predictions by normalizing meteorological factors. This would give the third line in each panel of figure 3. You can do this by following the methodology in figure 5 of Grange et al 2019 and figure 2 of Yang et al 2021. Vu et al 2019 made an additional improvement to weather normalization which you may also consider using this methodology. | We decided to adopt this suggestion making the following changes to our study:

 (1) Building a new RF model, normalizing the weather variables, adopting the Shi et al 2021 approach (which follows Vu et al 2019 not normalizing time variables, and Grange et al 2019 resampling from the whole study period). With this new model, we re-estimate the relative changes between MAM 2019 and MAM 2020, but using the normalized concentrations. This new approach allowed us to re-analyze the effects of the emissions changes during COVID-19 period, decoupling the effects of meteorological conditions. These new results will be included in the revised version of the manuscript.

 (2) Leaving the previous RF model as a predictive tool for air quality forecast in Buenos Aires, but adding new explanatory variables, as explained in Answer #1. This approach allowed us to assess the combined effect of the particular meteorological situation during COVID-19 restriction period, and the reduction in the emissions that actually occurred. The comparison between observations and concentrations that would have occurred under normal emissions conditions (BAU scenario) estimated with this RF model will be kept in the revised version of the manuscript.

 To explain these methodological changes Figure 2 in the revised version will be replaced by Figure 2 (revised) presented further down.

 In addition, we will expand our references, including Le et al, 2020, Vu et al 2019, Shi et al 2021 and Grange et al 2019 |

[Figure]

Figure 2 (revised)

3. The description of details of why and how to interpret bivariate polar plots here is vague. The reason that Grange 2019 used this bivariate polar plot is that they wanted to prove that wind influences the dispersion of pollutants. Therefore it's better to show the meteorological impacts from the above suggestions that I mentioned first. Then applying a bivariate polar plot by combining with winds components if the meteorological impact is the dominant factor here.

Even though we knew that wind speed and direction were key variables for pollutant dispersion in Buenos Aires, we used bivariate plots not for proving that, but as a tool for source location, detection and characterisation (following Grange et al 2016 and Carslaw 2013). We think that these graphs provide a graphical support for showing joint wind speed, wind direction dependence of air pollutant concentrations under different scenarios, in this case with and without COVID-19 restrictions. We used these plots to gain knowledge on potential sources that were missing during COVID-19 lockdown periods, such as the impact on the different pollutants of reducing vehicular emissions.

In order to improve the way to interpret these useful plots, we will modify the line 228, adding the following sentence at the end: "These should help us detect, locate and characterize pollution sources, as done by Carslaw *et al.* 2013 and by

| | | Grange *et al.* 2016." |
|---|---|---|
| | | *Stuart K. Grange, Alastair C. Lewis, David C. Carslaw, Source apportionment advances using polar plots of bivariate correlation and regression statistics, Atmospheric Environment, Volume 145, 2016.*
 *Carslaw 2013, David C. Carslaw, Sean D. Beevers, Characterizing and understanding emission sources using bivariate polar plots and k-means clustering* |
| 4. | For the relationship between $NO_2$ and diesel please refer to Yang et al 2021. You can also consider normalizing other anthropogenic factors besides meteorology based on the result from the first suggestion. | Thanks for your suggestion. Unfortunately we couldn't consider other anthropogenic factors because there is no available traffic data in Buenos Aires, with the temporal and geographical disaggregation needed. |
| 5. | Please indicate how much data by # not percentage is used for training and how much data is used for validation/prediction. Due to some restrictions of the monitoring campaign, please indicate how the authors dealt with inadequate data for specific variables. | The CNEA site has 9198 valid data points but only 7150 of those were before the BLD period (please, refer to Figure 2 revised for period definitions); 80% of these (5720) were used to train the model and the rest (1430) for testing. The independent period for evaluation (namely BLD) is composed of 360 data points. There are 4 days between BLP and LD that were not taken into account, because they have been considered as a "transition period". The numbers for the Parque Centenario site are: 8710 data points before the BLD period, with 6968 used for training and the rest for testing (1742).

 Inadequate data was considered as missing data, and was not replaced.

 The revised version of the manuscript will clarify this issue, modifying the lines 188-192. |
| 6. | The clarity and context need significant improvement to better draw out why the results are significant. | We tried to emphasize in the conclusions the importance of our results, including:
 A. The importance of producing novel $SO_2$ and $O_3$ data, in a basin with lack of monitoring data for these pollutants in residential/commercial areas (the only data available corresponds to an industrial area). It is well-known the importance of having non-industrial air quality data for air quality model validation.
 B. The importance of having a tool for air quality forecasts in Buenos Aires at a low computational cost, which could be useful for air quality management in the city.
 C. The analysis of the effects of COVID restrictions in air quality, analyzing (as was made in the rest of the world) the effects in |

| | |
|---|---|
| | primary pollutants reduction, but also in $O_3$ increase. |
| | D. The probable VOCs limited regime for the Buenos Aires Atmosphere. |
| | As was said in our Answer#1, the revised version of our work will include a change in the abstract reflecting all these issues , but we will also include modifications in the conclusions to better highlight the relevance of our results. |
| The following paper should have been referenced and discussed in the manuscript: Grange, Stuart K., and David C. Carslaw. "Using meteorological normalisation to detect interventions in air quality time series." Science of the Total Environment 653 (2019): 578-588. Yang, Jiani, Yifan Wen, Yuan Wang, Shaojun Zhang, Joseph P. Pinto, Elyse A. Pennington, Zhou Wang et al. "From COVID-19 to future electrification: Assessing traffic impacts on air quality by a machine-learning model." Proceedings of the National Academy of Sciences 118, no. 26 (2021). Vu, Tuan V., Zongbo Shi, Jing Cheng, Qiang Zhang, Kebin He, Shuxiao Wang, and Roy M. Harrison. "Assessing the impact of clean air action on air quality trends in Beijing using a machine learning technique." Atmospheric Chemistry and Physics 19, no. 17 (2019): 11303-11314. Le, Tianhao, Yuan Wang, Lang Liu, Jiani Yang, Yuk L. Yung, Guohui Li, and John H. Seinfeld. "Unexpected air pollution with marked emission reductions during the COVID-19 outbreak in China." Science 369, no. 6504 (2020): 702-706. | Thank you for this suggestion, we will add these references to the manuscript. |

---

## Author Comment (AC2)

Answer Referee #2

**General comments to both referees:**

Most of the suggestions received have been adopted. From this, major changes have been made on the study, comprising: (1) the inclusion of Gini importance plots; (2) a modification of the set of predictive variables, adding the boundary layer height and the total cloud cover and (3) the normalization of the meteorological variables. On this basis, we build a new model to decouple the effect of the meteorology for the analysis of the relative changes during COVID-19 period. Although these suggestions led to an overall better implementation of the Random Forest model and a better subsequent analysis, the conclusions remain almost unchanged. Nevertheless, with the new estimated values for the relative changes, we will include substantive modifications in section 3.1.

| | |
|---|---|
| The manuscript is generally well written and clearly presented. However, its research outcome (i.e the impact of meteorology and regional sources on air quality in Buenos Aires, Argentina) is not new. It should investigate the interactions between input variables to understand more about the Random forest model. I do recommend publishing this work if the authors can solve my major concerns as below: | |
| Major concerns:

1.  Selection of explanatory variables:

1.1. Table 3, line 200-205: Air quality strongly depends upon boundary layer height and long-range transport. Why were these variables not included in this study as input variables in the model? Please refer to a reference by Shi et al. 2021 (Science Advance, Vol 7, Issue 3, "Abrupt but smaller than expected changes in surface air quality attributed to Covid-19 lockdowns"). | Thank you for this recommendation. Following your suggestion we added the boundary layer height as an explanatory variable, and also explored (and finally included) other explanatory variables that were used by Shi et al. 2021, such as total cloud cover.
Regarding the intrusion of regional plumes, in general terms, in Buenos Aires local sources prevailed over regional ones, as was also seen in Diaz Resquin et al 2018. Particularly, during the study period (2019 to March 2020) we have been working on the identification of these events undertaking a detailed (day by day) analysis of satellite images and derived products. From this work, which has not yet been published, we found about 10 days with regional plumes impacting at ground level. Supporting that, Otero et al 2020 showed the existence of one of these situations (November 2019 and January 2020) under which regional plumes from Australia fires reached the city, but passed above the boundary layer height. Taking into account all these results, we decided to disregard the long-range transport in this first version of the model. Nevertheless we agree with you that this model could not be representative of other particular periods, where for example the impact of biomass burning events is relevant in terms of the City's air quality. We will consider this variable for future model improvements. |
| 1.2. Could the author explain why | Several model architectures have been explored |

| | |
|---|---|
| the cho CO, NO as explanatory variables for $NO_2$? CO and NO were modeled from t2, rh2, U, V and gasoline diurnal patterns, so I guess the author also can model $NO_2$ based on these variables. Similar questions for explanatory variables for $SO_2$ and $PM_{10}$, and $O_3$. | before choosing the final explanatory variables set. We started using the meteorological and emission patterns variables for all the pollutants analyzed, but found better model results with the configuration presented in the manuscript. The goodness of this selection has been revealed in the Gini Index Figures prepared during this review (see Answer #1 to Reviewer #1). As an example, please take a look of Gini Index for $O_3$ using all the explanatory variables:

[Figure]
 |
| 1.3. In the model of $NO_2$, did the author investigate interactions between input variables such as NO with t2. | The interactions between input variables have been analyzed estimating the partial dependence between the variables (using the rmweather R package), but also from correlations plots. Part of the analysis performed was included in sections 3.2.1 to 3.2.5. As part of this analysis, we investigated the interactions between NO and t2 in the model of NO2, obtaining the expected behavior (i.e. high temperatures favor the conversion of NO to $NO_2$). In the revised version of the manuscript we will expand section 3.2.2, including the role of NO and t2 in the model of $NO_2$. |
| 1.4. In terms of $O_3$, it strongly depends upon atmospheric temperature. Why does this variable not be included in your model? | We agree with you about the existence of a strong relationship between $O_3$ and t2. It has also been raised in the partial dependence plots. On the revised version of the model (see Answer#1 to Reviewer#1) t2 has been included as an explanatory variable for $O_3$ . |
| 2. Testing dataset: | Answer to Q 2.1: |

| | |
|---|---|
| 2.1. Figure 2, Line 189: What criteria do authors select testing dataset based on (i,e 2 weeks data before lockdown?)

2.2. In my opinion, the 2-weeks data for testing data sets is too short. Therefore, authors should do a model performance for at least one month before and after the lockdown/partial periods. | We adopted a similar approach applied by Grange et al 2021: (1) all the data measured from Feb-23-2019 to Feb-15-2020 has been randomly split between training (80%) and testing (20%); (2) in addition, to check the adequate model performance under BAU scenario, we used an independent evaluation period two weeks before lockdown. This issue will be clarified, modifying the Figure 2 (see Answer #2 Rev#1) and also the text from lines 187-189.

Answer to Q 2.2:

As was highlighted in the previous paragraph, the testing data set is 20% of almost 1 year (1430 data points out of 7150 for CNEA and 1742 out of 8710 for PC).
In relation with the independent evaluation period, originally we wanted to use 1 month before and 1 month after LD/PLD periods, but: (1) during the last two weeks of February the equipment in CNEA was out of service and (2) since access to the CNEA measuring station was strongly restricted, due to maintenance difficulties the equipment was turned off by May 2020 (line 158-159). |
| Minor comments:

3.1. Table 4: I think author should include the r value between model and observation rather the r-value for diurnal cycle (r-dc) | Answer to Q 3.1:

We included the correlation coefficient of the diurnal cycle because having an adjusted diurnal cycle is a major concern for the region. However, as the analysis of the diurnal cycle has been deeply discussed in the manuscript (Figures 3 and 4), we agree with you that the r value between model and observations will enhance the transparency of the results. Therefore, in the reviewed version of the manuscript, we will modify Table 4, replacing $r_{dc}$ by the r values between model and observations. |
| 3.2. Table 5: In BLD, it should include concentrations of pollutants between observation and model. | Answer to Q 3.2:

Thank you for this observation; we will modify the manuscript adding a new column in Table 5. |
| 3.3. In discussion: Authors should plot the dependence of concentration of pollutants on meteorological conditions. | As mentioned in Answer 1.3, in the revised version of the manuscript, we will include partial dependence plots in the supplementary material. In addition, in Section 3.1, we will add the analysis of these partial dependencies in depth. |

---

## Author Response (AR1)

Answer Referee #1

**General comments to both referees:**

Most of the suggestions received have been adopted. From this, major changes have been made on the study, comprising: (1) the inclusion of variable importance plots; (2) a modification of the set of predictive variables, adding the boundary layer height and the total cloud cover and (3) the normalization of the meteorological variables, which required substantive changes in section 2.5. On this basis, we implemented a normalization technique to decouple the effect of the meteorology for the analysis of the relative changes during COVID-19 period. Although these suggestions led to an overall better implementation of the Random Forest model and a better subsequent analysis, the conclusions remain almost unchanged. Nevertheless, with the new estimated values for the relative changes, we included substantive modifications in section 3.2.
* * *
This study was majorly inspired by Grange et al 2019 to use a random forest model (RF) to train, validate and predict the air quality concentration in a megacity of Argentina. Although the methodology is not new, this study has the potential to have a significant improvement to fill the data gap given the reason that some monitored trace gas concentrations were lacking but the pollutants are becoming a concern for local authorities. (line 145- 150) I would like to propose a few major revisions for the authors of this manuscript. After done with that improvement, I think it would be strong enough to publish on earth system science data:
* * *
1. Random forest models indeed are easy and quick to train. But if this study is only focusing on predicting the time series of pollution during COVID-19, there are better and more efficient machine learning models such as ARIMA compared with random forest. The key reason for many studies that authors mentioned in this article used RF is because it could provide key components reflecting the non-linear relationship among emissions, chemical reaction, and meteorological effects. Please see figure 3 in Grange et al 2019 and figure 1 in Yang et al. It is easy to generate that Gini importance either through python or R. Since this code the authors used were based on R, I would suggest they follow the code from Grange et al 2019 to generate the Gini

Following your suggestion, we generated the variable importance analysis using the *ranger* package in R for all the variables considered in the model. As you mentioned, these results were helpful for us to understand the underlying role (non-linear effects) of some variables in the concentrations of the pollutants analyzed. As an example, variable importance plots revealed the importance of considering the boundary layer height as an explanatory variable, particularly for CO and NO description. On the basis of this analysis, we modified the set of the explanatory variables chosen for each model, adding the boundary layer height (blh). For example, for the CNEA site: (1) for CO the set of explanatory variables was changed from {t2; rh2; U; V; gasoline diurnal cycle} to { t2; ws; wd; blh; gasoline diurnal cycle} and (2) for NO the set was modified from {t2; rh2; U; V; gasoline and diesel diurnal cycles} to {t2; rh2; sea level pressure; ws; wd; blh; total cloud cover; gasoline and diesel diurnal cycle}

Regarding the aim of the study, note that we presented 3 different goals. Two of them (lines 65-68 from the original manuscript) are: (1) the prediction of the time series during COVID-19 and (2) the development of a model for air quality forecasts for the Metropolitan Area of Buenos Aires at a low computational cost. In line 78 and 79, we also explain that we are providing (3) the first $O_3$ and $SO_2$ observational datasets in more than a decade in

| | |
|---|---|
| importance plot. After getting those plots, you can compare your RF results with the reasons from previous literature to see if it makes sense. | Buenos Aires. However, from your comment, we realized that (2) and (3) were not highlighted in the abstract. Therefore, we have included substantive modifications in the abstract. We also added there the analysis that we did including answer 2.

We also added a reference to Yang et al. 2021 and Grange et al 2019. |
| 2. The authors also mentioned several times the meteorological impacts on air pollution. The earliest signal has been seen during COVID was the study in China from Le et al, 2020 which authors may consider mentioning this study result. Then it would also be beneficial to consider using RF models to generate new predictions by normalizing meteorological factors. This would give the third line in each panel of figure 3. You can do this by following the methodology in figure 5 of Grange et al 2019 and figure 2 of Yang et al 2021. Vu et al 2019 made an additional improvement to weather normalization which you may also consider using this methodology. | We decided to adopt this suggestion making the following changes to our study:

(1) Building a new RF model, normalizing the weather variables, adopting the Shi et al 2021 approach (which follows Vu et al 2019 not normalizing time variables, and Grange et al 2019 resampling from the whole study period). With this new model, we re-estimate the relative changes between MAM 2019 and MAM 2020, but using the normalized concentrations. This new approach allowed us to re-analyze the effects of the emissions changes during COVID-19 period, decoupling the effects of meteorological conditions. These new results were included in the revised version of the manuscript.

(2) Leaving the previous RF model as a predictive tool for air quality forecast in Buenos Aires, but adding new explanatory variables, as explained in Answer #1. This approach allowed us to assess the combined effect of the particular meteorological situation during COVID-19 restriction period, and the reduction in the emissions that actually occurred. The comparison between observations and concentrations that would have occurred under normal emissions conditions (BAU scenario) estimated with this RF model was kept in the revised version of the manuscript.

To explain these methodological changes Figure 2 in the revised version was replaced by Figure 2 (revised) presented further down.

In addition, we will expand our references, including Le et al, 2020, Vu et al 2019, Shi et al 2021 and Grange et al 2019 |

[Figure]

Figure 2 (revised)

| 3. The description of details of why and how to interpret bivariate polar plots here is vague. The reason that Grange 2019 used this bivariate polar plot is that they wanted to prove that wind influences the dispersion of pollutants. Therefore it's better to show the meteorological impacts from the above suggestions that I mentioned first. Then applying a bivariate polar plot by combining with winds components if the meteorological impact is the dominant factor here. | Even though we knew that wind speed and direction were key variables for pollutant dispersion in Buenos Aires, we used bivariate plots not for proving that, but as a tool for source location, detection and characterisation (following Grange et al 2016 and Carslaw 2013). We think that these graphs provide a graphical support for showing joint wind speed, wind direction dependence of air pollutant concentrations under different scenarios, in this case with and without COVID-19 restrictions. We used these plots to gain knowledge on potential sources that were missing during COVID-19 lockdown periods, such as the impact on the different pollutants of reducing vehicular emissions.

In order to improve the way to interpret these useful plots, we added the following statement: "To detect, locate and characterize different pollution sources (Carslaw and Beevers, 2013; Grange et al., 2016), bivariate polar plots were built considering observations and RF results, (...)" |
|---|---|

| | |
|---|---|
| | *Stuart K. Grange, Alastair C. Lewis, David C. Carslaw, Source apportionment advances using polar plots of bivariate correlation and regression statistics, Atmospheric Environment, Volume 145, 2016.*
*Carslaw 2013, David C. Carslaw, Sean D. Beevers, Characterizing and understanding emission sources using bivariate polar plots and k-means clustering* |
| 4. For the relationship between $NO_2$ and diesel please refer to Yang et al 2021. You can also consider normalizing other anthropogenic factors besides meteorology based on the result from the first suggestion. | Thanks for your suggestion. Unfortunately we couldn't consider other anthropogenic factors because there is no available traffic data in Buenos Aires, with the temporal and geographical disaggregation needed. We now mentioned this limitations explicitly |
| 5. Please indicate how much data by # not percentage is used for training and how much data is used for validation/prediction. Due to some restrictions of the monitoring campaign, please indicate how the authors dealt with inadequate data for specific variables. | The PC site had 9198 valid data points but only 7150 of those were before the BLD period (please, refer to Figure 2 revised for period definitions); 80% of these (5720) were used to train the model and the rest (1430) for testing. The independent period for evaluation (namely BLD) was composed of 360 data points. There are 4 days between BLP and LD that were not taken into account, because they have been considered as a "transition period".
The numbers for the CNEA site were: 8710 data points before the BLD period, with 6968 used for training and the rest for testing (1742).

Inadequate data was considered as missing data, and was not replaced.

The revised version of the manuscript clarifies this issue in section 2.5. |
| 6. The clarity and context need significant improvement to better draw out why the results are significant. | We tried to emphasize in the conclusions the importance of our results, including:
A. The importance of producing novel $SO_2$ and $O_3$ data, in a basin with lack of monitoring data for these pollutants in residential/commercial areas (the only data available corresponds to an industrial area). It is well-known the importance of having non-industrial air quality data for air quality model validation.
B. The importance of having a tool for air quality forecasts in Buenos Aires at a low computational cost, which could be useful for air quality management in the city.
C. The analysis of the effects of COVID restrictions in air quality, analyzing (as was made in the rest of the world) the effects in primary pollutants reduction, but also in $O_3$ increase.
D. The probable VOCs limited regime for the Buenos |

| | |
|---|---|
| | Aires Atmosphere.
As was said in our Answer#1, the revised version of our work includes a change in the abstract reflecting all these issues , but we also included modifications in the conclusions to better highlight the relevance of our results. Some conclusions were added about the usefulness of the meteorological normalization. |
| The following paper should have been referenced and discussed in the manuscript: Grange, Stuart K., and David C. Carslaw. "Using meteorological normalisation to detect interventions in air quality time series." Science of the Total Environment 653 (2019): 578-588.
Yang, Jiani, Yifan Wen, Yuan Wang, Shaojun Zhang, Joseph P. Pinto, Elyse A. Pennington, Zhou Wang et al. "From COVID-19 to future electrification: Assessing traffic impacts on air quality by a machine-learning model." Proceedings of the National Academy of Sciences 118, no. 26 (2021).
Vu, Tuan V., Zongbo Shi, Jing Cheng, Qiang Zhang, Kebin He, Shuxiao Wang, and Roy M. Harrison. "Assessing the impact of clean air action on air quality trends in Beijing using a machine learning technique." Atmospheric Chemistry and Physics 19, no. 17 (2019): 11303-11314.
Le, Tianhao, Yuan Wang, Lang Liu, Jiani Yang, Yuk L. Yung, Guohui Li, and John H. Seinfeld. "Unexpected air pollution with marked emission reductions during the COVID-19 outbreak in China." Science 369, no. 6504 (2020): 702-706. | Thank you for this suggestion, we added these references to the manuscript. |

Answer Referee #2

**General comments to both referees:**
Most of the suggestions received have been adopted. From this, major changes have been made on the study, comprising: (1) the inclusion of variable importance plots; (2) a modification of the set of predictive variables, adding the boundary layer height and the total cloud cover and (3) the normalization of the meteorological variables, which required substantive changes in section 2.5. On this basis, we implemented a normalization technique to decouple the effect of the meteorology for the analysis of the relative changes during COVID-19 period. Although these suggestions led to an overall better implementation of the Random Forest model and a better subsequent analysis, the conclusions remain almost unchanged. Nevertheless, with the new estimated values for the relative changes, we included substantive modifications in section 3.2.

| |
|---|
| The manuscript is generally well written and clearly presented. However, its research outcome (i.e the impact of meteorology and regional sources on air quality in Buenos Aires, Argentina) is not new. It should investigate the interactions between input variables to understand more about the Random forest model. I do recommend publishing this work if the authors can solve my major concerns as below: |

| Major concerns: | Thank you for this recommendation. Following your suggestion we added the boundary layer height as an explanatory variable, and also explored (and finally included) other explanatory variables that were used by Shi et al. 2021, such as total cloud cover. |
|---|---|
| 1. Selection of explanatory variables:

1.1. Table 3, line 200-205: Air quality strongly depends upon boundary layer height and long-range transport. Why were these variables not included in this study as input variables in the model? Please refer to a reference by Shi et al. 2021 (Science Advance, Vol 7, Issue 3, "Abrupt but smaller than expected changes in surface air quality attributed to Covid-19 lockdowns"). | Regarding the intrusion of regional plumes, in general terms, in Buenos Aires local sources prevailed over regional ones, as was also seen in Diaz Resquin et al 2018. Particularly, during the study period (2019 to March 2020) we have been working on the identification of these events undertaking a detailed (day by day) analysis of satellite images and derived products. From this work, which has not yet been published, we found about 10 days with regional plumes impacting at ground level. Supporting that, Otero et al 2020 showed the existence of one of these situations (November 2019 and January 2020) under which regional plumes from Australia fires reached the city, but passed above the boundary layer height. Taking into account all these results, we decided to disregard the long-range transport in this first version of the model. Nevertheless we agree with you that this model could not be representative of other particular periods, where for example the impact of biomass burning events is relevant in terms of the City's air quality. We will consider this variable for future model improvements. |

| | |
|---|---|
| 1.2. Could the author explain why the cho CO, NO as explanatory variables for $NO_2$? CO and NO were modeled from t2, rh2, U, V and gasoline diurnal patterns, so I guess the author also can model $NO_2$ based on these variables. Similar questions for explanatory variables for $SO_2$ and $PM_{10}$, and $O_3$. | Previously, we had tried several model architectures before choosing the final explanatory variables set. In this reviewed manuscript, we added some discussion about that (section 2.5), and explained the choice of the set of parameters according to the performance during testing. The goodness of this selection has been revealed in the Variable importance plots prepared during this review (see Answer #1 to Reviewer #1). As an example, please take a look of variable importance plot for $O_3$ using all the explanatory variables:
[Figure]
 |
| 1.3. In the model of $NO_2$, did the author investigate interactions between input variables such as NO with t2. | The interactions between input variables have been analyzed estimating the partial dependence between the variables (using the rmweather R package), but also from correlations plots. Part of the analysis performed was included in sections 3.2.1 to 3.2.5. As part of this analysis, we investigated the interactions between NO and t2 in the model of NO2, obtaining the expected behavior (i.e. high temperatures favor the conversion of NO to $NO_2$). In the revised version of the manuscript we expanded section 3.1, including the role of NO and t2 in the model of $NO_2$. |
| 1.4. In terms of $O_3$, it strongly depends upon atmospheric temperature. Why does this variable not be included in your model? | We agree with you about the existence of a strong relationship between $O_3$ and t2. It has also been raised in the partial dependence plots. In spite of this, the inclusion of t2 as explanatory variable worsened the performance of the model. However, as we discussed in section 3.2.3, these effects were indirectly included through its chemical precursors (see new version of Table 3). |
| 2.  Testing dataset:

2.1. Figure 2, Line 189: What criteria do authors select testing dataset based on (i,e 2 weeks data before lockdown?)

2.2. In my opinion, the 2-weeks data for testing data sets is too short. | Answer to Q 2.1:

We adopted a similar approach applied by Grange et al 2021: (1) all the data measured from Feb-23-2019 to Feb-15-2020 has been randomly split between training (80%) and testing (20%); (2) in addition, to check the adequate model performance under BAU scenario, we used an independent evaluation period two weeks before lockdown. This issue was clarified, |

| | |
|---|---|
| Therefore, authors should do a model performance for at least one month before and after the lockdown/partial periods. | modifying the Figure 2 (see Answer #2 Rev#1) and also the text from section 2.5.

Answer to Q 2.2:

As was highlighted in the previous paragraph, the testing data set is 20% of almost 1 year (1430 data points out of 7150 for CNEA and 1742 out of 8710 for PC).
In relation with the independent evaluation period, originally we wanted to use 1 month before and 1 month after LD/PLD periods, but: (1) during the last two weeks of February the equipment in CNEA was out of service and (2) since access to the CNEA measuring station was strongly restricted, due to maintenance difficulties the equipment was turned off by May 2020 (line 178-179 from the revised manuscript). |
| Minor comments:

3.1. Table 4: I think author should include the r value between model and observation rather the r-value for diurnal cycle (r-dc)

3.2. Table 5: In BLD, it should include concentrations of pollutants between observation and model. | Answer to Q 3.1:

We included the correlation coefficient of the diurnal cycle because having an adjusted diurnal cycle is a major concern for the region. However, as the analysis of the diurnal cycle has been deeply discussed in the manuscript (Figures 3 and 4), we agree with you that the r value between model and observations will enhance the transparency of the results. Therefore, in the reviewed version of the manuscript, we modified Table 4, replacing $r_{dc}$ by the r values between model and observations.

Answer to Q 3.2:

Thank you for this observation; we modified the manuscript adding new columns to Table 5. |
| 3.3. In discussion: Authors should plot the dependence of concentration of pollutants on meteorological conditions. | As mentioned in Answer 1.3, in the revised version of the manuscript, we included partial dependence plots in the supplementary material. In addition, in Section 3.1, we added the analysis of these partial dependencies. |

**List of the relevant changes:**

- Substantive changes: (1) Section 2.5, where the new modeling approach is described, including a new set of predictive variables and a normalization technique for the meteorological variables; (2) Section 3.2, where the analysis of the normalized concentrations was added (obtaining similar results of the analysis presented in the previous version); (3) Section 3.2.4, where the partial dependencies plots, included in the new version (thanks to Reviewer#1's suggestion) enlighten the role of shipping as a relevant $SO_2$ emission source in Buenos Aires.

- The Abstract was rewritten to better reflect the relevance and goals of the work performed.
- Several minor changes were also included, mostly related with nomenclature modifications.

---

## Referee Report (RR1)

The authors did a great job in improving the quality of this manuscript compared with the previous version. The simulating results of different pollutants are well explained e.g. $NO_2$ is partially secondary in the primary pollutant of NO, the night reaction between $O_3$ and $NO_2$, and the photochemistry of $O_3$ under different regimes. I would recommend the editor accept this manuscript after minor revisions as below:

- Line 250: please add more interpretation on how to calculate the partial dependencies in Figs. S9 and S11.
- Line 265: In Fig.S4, there is no $O_3$-CO ratio from the plot (red dots are missing) and please add it on.
- Please mark the title of x-axis of Fig.3.
- Please mark the title of x-axis of Fig.S9.
- Please mark the title of the x and y axis of Fig.S9 and Fig.S11.
- Please indicate which year it was for Fig.S5.
- The authors did a great job to export the variable importance plots (Fig. S6 – S8) but it was not mentioned in the main text. It would be great to add those importance plots analysis in section 3.2.

---

## Author Response (AR2)

Dear Topical Editor,

We welcome all the comments and suggestions made, which have helped us improve our manuscript. We trust that we have responded satisfactorily to all your comments in the document attached to this response.

| | |
|---|---|
| Line 250: please add more interpretation on how to calculate the partial dependencies in Figs. S9 and S11. | Lines from 249 to 253 were added:

"Partial dependencies plots were also built using the rmweather library of R (Grange et al., 2018; Grange and Carslaw, 2019) to highlight the relationships between pollutant concentrations and all explanatory variables presented in Table 3, and can be seen in the Supplementary Information (Figs. S9 to S11). By obtaining the prediction from the random forest model for each unique value of a specific explanatory variable, these plots allow us to analyze how this dependency varies for different values of the explanatory variable, and therefore help us to detect non-linear relationships, which are highly relevant in air quality." |
| Line 265: In Fig.S4, there is no O3-CO ratio from the plot (red dots are missing) and please add it on. | Thank you for pointing this out, they were out of scale, the plot was corrected. |
| Please mark the title of x-axis of Fig.3. | Figures were corrected as you suggested and also Figs. 6 and 7 where the x-axis was also missing. |
| Please mark the title of x-axis of Fig.S9. | |
| Please mark the title of the x and y axis of Fig.S9 and Fig.S11. | |
| Please indicate which year it was for Fig.S5. | The legend was modified to add the period for that figure: "during the training period (February 2019 to February 2020)" |
| The authors did a great job to export the variable importance plots (Fig. S6 – S8) but it was not mentioned in the main text. It would be great to add those importance plots analysis in section 3.2. | As Section 3.2 was focused on assessing the lockdown changes, we expanded the variable importance plots analysis in Section 3.1 where the general analysis of the random forest model was done (Lines 279-287). Also Figure 5 was added with the Variable importance for CO. |